# AutoEdit: Automatic Hyperparameter Tuning for Image Editing

Chau Pham[1]    Quan Dao[2]    Mahesh Bhosale[1]    Yunjie Tian[1]*

Dimitris Metaxas[2]    David Doermann[1]

[1]University at Buffalo
[2]Rutgers University

## Abstract

Recent advances in diffusion models have revolutionized text-guided image editing, yet existing editing methods face critical challenges in hyperparameter identification. To get the reasonable editing performance, these methods often require the user to brute-force tune multiple interdependent hyperparameters, such as inversion timesteps and attention modification, *etc.* This process incurs high computational costs due to the huge hyperparameter search space. We consider searching optimal editing's hyperparameters as a sequential decision-making task within the diffusion denoising process. Specifically, we propose a reinforcement learning framework, which establishes a Markov Decision Process that dynamically adjusts hyperparameters across denoising steps, integrating editing objectives into a reward function. The method achieves time efficiency through proximal policy optimization while maintaining optimal hyperparameter configurations. Experiments demonstrate significant reduction in search time and computational overhead compared to existing brute-force approaches, advancing the practical deployment of a diffusion-based image editing framework in the real world. Codes can be found at `https://github.com/chaupham1709/AutoEdit.git`.

## 1 Introduction

Image generation has recently witnessed remarkable advancements through diffusion models [13, 34, 3, 32, 36], driving a growing interest in their broad applicability. Within this domain, prompt-to-prompt image editing [12, 5, 4, 16, 45, 26, 14, 8] has emerged as a critical subfield focused on modifying image content according to textual instructions. This task focuses on achieving two essential objectives: (1) precisely aligning the modified image with the editing prompt, while (2) minimizing unnecessary alterations to the original content beyond those required by the editing instruction. Maintaining this delicate balance between instruction alignment and background preservation is the primary challenge.

Despite notable advancements, current image editing methods [5, 12, 39, 15, 16, 17] suffer from a heavy reliance on manual hyperparameter identification during the editing process. These hyperparameters span multiple dimensions: inversion timestep configuration [39, 15, 11], attention modulation mechanisms [12, 16] (including layer-wise modification and semantic reweighting), and feature blending coefficients [17, 11]. As empirically demonstrated in [45], the optimal hyperparameter combination exhibits strong sensitivity across different images. This requires exhaustive trial-and-error iterations to identify optimal hyperparameter sets that simultaneously satisfy two

---

*Corresponding author

39th Conference on Neural Information Processing Systems (NeurIPS 2025).

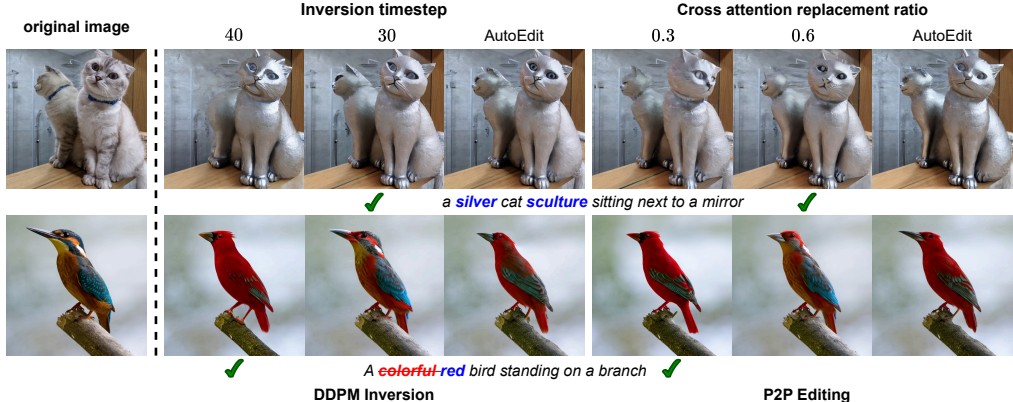

Figure 1: **Optimal hyperparameters vary significantly across images**: The cat image achieves best editing at step 30 while the bird requires step 40 in timestep experiments, with similar variance observed in P2P [12] cross-attention ratios. Our AutoEdit automatically identifies near-optimal configurations across these parameters, matching manual search performance (last column).

criteria: (1) achieving instruction-aligned editing results while (2) preserving original background. Such a hyperparameter search introduces significant computational overhead and creates substantial usability barriers for non-expert users, as evidenced by our quantitative illustration in Figure 1.

Previous work OIR [45] optimize the value of hyperparameter "inversion timestep $r$", which is the maximum timestep we try to invert image in editing process, for each pair (input images, edit prompt) to obtain better editing result. Specifically, OIR proposes a brute-force search strategy that evaluates all possible $r \in \{1, 2, \ldots, T\}$ ($T$ is total denoising step) and selects the optimal $r^*$ to achieve the highest editing score. Furthermore, this approach requires $T$ denoising processes per editing sample, which consume $\mathcal{O}(T^2)$ number of functional evaluation (NFEs). This method becomes impractical when $T$ is large, which is around 50 or 100 in case of editing process. However, OIR only focuses on choosing hyperparameter $r$ while neglecting other critical hyperparameters.

For each editing method, we define a hyperparameter set $\mathcal{H} = \{h^1, h^2, ..., h^K\}$, consisting of $K$ distinct hyperparameters. Assuming each $h^k$ can take on $N$ possible values, a brute-force search for the optimal configuration incurs a computational complexity of $\mathcal{O}(TN^K)$, which is impractical in real-world scenarios. To address this, we propose an efficient hyperparameter identification framework that reduces the search complexity from exponential $\mathcal{O}(TN^K)$ to linear $\mathcal{O}(T)$. Conceptually, a diffusion-based editing process can be viewed as a denoising procedure influenced by the hyperparameter set $\mathcal{H}$. This perspective allows us to formulate the editing process as RL-driven Markov Decision Process (MDP). Specifically, each denoising step $t = T \rightarrow 1$ corresponds to a state $s_t = (x_t, t)$, where $x_t$ is the noisy latent at timestep $t$. We treat each hyperparameter $h^k$ as a sequence $\{h_t^k\}_{t=T \rightarrow 1}$, and collectively, $\mathcal{H}_t = \{h_t^k\}_{k=1}^K$ represents a set of parallel actions taken at state $s_t$. The reward function is designed to integrate two essential editing criteria: prompt alignment and background preservation. Optimization is performed using Proximal Policy Optimization (PPO) [37]. At each timestep $t$, the learned policy selects the optimal hyperparameter set $\mathcal{H}_t$, which is directly applied during editing process. As the result, our technique could save the user from heuristically choosing hyperparamenter.

Our core contributions are threefold: (1) First formulating optimal hyperparameter identification as a key challenge in diffusion-based editing, exposing the computational bottleneck $\mathcal{O}(TN^k)$ of brute force search; (2) A reinforcement learning framework that unifies hyperparameter identification with standard denoising, achieving efficient $\mathcal{O}(T)$ optimal hyperparameter search; (3) Empirical validation showing that our method nearly achieves optimal hyperparameter while reducing search time by around three times versus brute-force baselines.

## 2 Related work

**Text-prompted Image Editing.** Text-based image editing [4, 5, 12, 8, 11, 40, 38, 31] modifies image content based on editing instructions, aiming to generate edited images that faithfully adhere to the

editing prompts while preserving most original background from the source image. While Generative Adversarial Networks (GANs) [27, 21, 43] achieved partial success in domain-specific datasets (e.g., facial images), they struggle with text-based editing due to low performance of text-to-image GAN. This limitation has been substantially addressed by pretrained text-to-image diffusion models [34], which have enabled the emergence of sophisticated editing frameworks [4, 5, 12, 8, 11, 40, 38, 31] through three principal paradigms: (1) prompt-to-prompt manipulation [5, 12, 15, 16, 17, 45], (2) instruction-based editing [4, 14, 31], and (3) image personalization [35, 9]. However, existing methods require brute-force search for optimal editing hyperparameters, which is a non-trivial process due to extensive trial-and-error iterations. Motivated from reinforcement learning in improving image generation [24, 10, 46, 29], we propose an RL framework for searching optimal hyperparameter in text-based image editing.

**RL in Image Generation.** Recent works have demonstrated reinforcement learning (RL) as a powerful paradigm for enhancing text-to-image generation [41, 23, 7, 29]. Pioneering works [24, 10, 46, 29] establish RL frameworks that optimize text prompts through iterative reward feedback, effectively aligning the generated images with the aesthetic score and semantic metric. By designing domain-specific reward mechanisms, these methods could effectively generate high quality images aligning with reward function. Inspired by this line of research but differently, we propose formulating an RL environment for the image editing problem to identify the optimal hyperparameters.

# 3 Method

## 3.1 Preliminaries

**Diffusion Models.** Diffusion Models [13, 28, 39, 34] generate images by iteratively denoising Gaussian noise. They consist of:

`Forward Process:` Add noise to a clean image $x_0$ over $T$ steps:

$$x_t \sim \mathcal{N}\big(\sqrt{\overline{\alpha}_t}\,x_0,\ (1 - \overline{\alpha}_t)\mathbf{I}\big),$$

with $\overline{\alpha}_t$ decreasing from 1 to 0, thus $x_T \sim \mathcal{N}(0, \mathbf{I})$.

`Reverse Process:` Learn backward transitions

$$p(x_{t-1}\,|\,x_t) = \mathcal{N}\big(x_{t-1}; \mu_\theta(x_t, t),\ \sigma_t^2\mathbf{I}\big),$$

where

$$\mu_\theta(x_t, t) = \frac{1}{\sqrt{\alpha_t}}\Big(x_t - \epsilon_\theta(x_t, t)\,\frac{1-\alpha_t}{\sqrt{1-\overline{\alpha}_t}}\Big), \quad \alpha_t = \frac{\overline{\alpha}_t}{\overline{\alpha}_{t-1}}.$$

The noise prediction network $\epsilon_\theta$ is trained by

$$\mathcal{L}_{\text{simple}} = \mathbb{E}_{x_0,\, \epsilon \sim \mathcal{N}(0,\mathbf{I}),\, t}\big\|\epsilon - \epsilon_\theta(x_t, t)\big\|_2^2.$$

**Image Editing.** Image editing [5, 12, 45, 17, 40, 1] modifies images to align with editing prompts while preserving background. Under the prompt-to-prompt edit setting, we denote the source image $I^{src}$, source prompt $p_{src}$, edit prompt $p_{edit}$, and edit region mask $M$.

The editing pipeline contains two stages: 1. **Inversion:** From source image $I^{src}$ and source prompt $p_{src}$, we extract noise set using an inversion technique. 2. **Denoising:** Using the editing prompt $p_{edit}$ and extracted noise set, we gradually denoise the edited image $I^{edit}$. During the above denoising stage, the users often need to **search for optimal hyperparameter** (e.g. attention weights, inversion steps) to obtain the edited image aligning to $p_{edit}$ and preserving background. Formally, for editing method $\mathcal{E}$ with hyperparameter set $\mathcal{H}$, we need to find

$$\mathcal{H}^* = \arg\min_{\mathcal{H}}\ \mathcal{D}\big(I^{edit}(\mathcal{E}(I^{src}, p_{edit}, \mathcal{H})),\ I^{target}\big),$$

where $\mathcal{D}$ measures how well $I^{edit}$ matches the edit criteria.

**Proximal Policy Optimization (PPO).** PPO [37] is a popular and reliable method in Reinforcement Learning, and is widely applied in LLM training [30], prompt optimization [24, 10]. PPO is a policy gradient method, where the policy model is updated by the expected gradient of the policy output and the advantage. The common technique in Policy Gradient is to train two seperate networks for

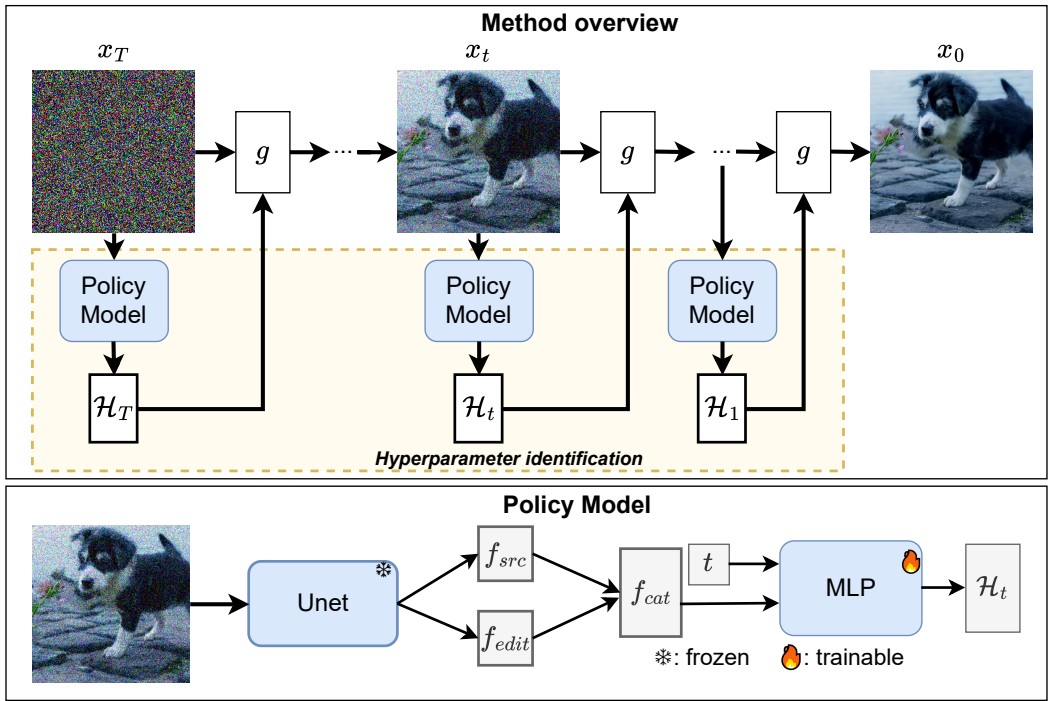

Figure 2: **Top**: Overview of the proposed AutoEdit framework. A policy model is injected to predict the step-wise hyperparameter $\mathcal{H}_t$ at each denoising step $t$. The predicted $\mathcal{H}_t$ is used with the one-step denoising function $g$ and current state $x_t$ to estimate $x_{t-1}$. **Bottom**: Architecture of the policy model. Features from the U-Net encoder under the original and edited prompts are extracted and concatenated, followed by several trainable layers to predict the policy output.

estimating the policy output $\pi_\theta$ (policy model) and the other network to estimate the reward to go $V_\tau$ (value model). In PPO, the policy model is updated by:

$$L(\theta) = E_t[\min(r_t A_t, \text{clip}(r_t, 1 - \epsilon, 1 + \epsilon) A_t)]$$

where $A_t$ is the advantage computed at time $t$ of the rollout process, and $r_t = \frac{\pi_\theta(a_t|s_t)}{\pi_{\theta_{old}}(a_t|s_t)}$ is the ratio between current and old policy. The value model is updated by:

$$L(\tau) = E_t[(V_\tau(s_t) - \hat{R}_t)^2]$$

where $\hat{R}_t$ is the expected return, which can be computed through the GAE method.

**Hyperparameter Selection.** Given specific $p_{edit}$ and $I^{src}$, editing performance heavily depend on hyperparameters $\mathcal{H}$. Naive trial-and-error tuning runs denoising for each $\mathcal{H}$, which is infeasible as $k$ grows. Previous methods like OIR [45] perform $T$ editing trials for $r \in \{1, 2, ..., T\}$, but (1) searching cost scales to $\mathcal{O}(T^2)$ NFEs and (2) they tune the only hyperparameter: the inversion timestep. We introduce AutoEdit, a single-pass denoising framework that efficiently finds $\mathcal{H}^*$.

### 3.2 AutoEdit: Automated Hyperparameter Identification

#### 3.2.1 RL Environment Formulation

We formulate the problem of optimizing hyperparameters as a RL task by integrating the RL framework into the diffusion denoising process. In this section, we define the key components of the RL environment:

**State.** In RL environment, we define $s_t = (x_t, t)$ as the state, with $x_t$ is noisy latent at time $t = T \to 1$. The initial state is $s_T = (x_T, T)$

**Action.** At each state $s_t$ at timestep $t$, selecting a hyperparameter configuration is treated as an action. Given a set of $K$ hyperparameters $\mathcal{H} = \{h^1, h^2, \ldots, h^K\}$, we aim to determine the optimal configuration $\mathcal{H}_t = \{h_t^k\}_{k=1}^K$ at every $s_t$ to maximize editing performance. This naturally leads us

to model each hyperparameter $h^k$ as a temporal sequence $\{h_t^k\}_{t=T\to 1}$ across the editing trajectory. As a concrete example, consider the case where $\mathcal{H} = \{r\}$, with $r$ denoting the inversion timestep. The editing process can be divided into two consecutive stages: denoising from $T$ to $r$ using the source prompt $p_{\text{src}}$, followed by denoising from $r$ to 1 using the edit prompt $p_{\text{edit}}$. In this formulation, choosing $r$ is equivalent to defining a prompt sequence $\mathcal{H}_t = \{h_t\}$, where $h_t \in \{p_{\text{src}}, p_{\text{edit}}\}$ depending on whether $t > r$ or $t \leq r$. Other hyperparameters can be similarly represented as time-dependent sequences similar the inversion timestep $r$, as elaborated in the Appendix. Therefore, at each timestep $t$, the configuration $\mathcal{H}_t$ constitutes a set of parallel actions over all $K$ hyperparameters at state $s_t$, enabling flexible and fine-grained control throughout the editing process. Once the hyperparameter configuration $\mathcal{H}_t$ is defined, the next state $x_{t-1}$ can be found by applying the diffusion denoising step $x_{t-1} = g(x_t, t, \mathcal{H}_t)$, where $g$ is one-step denoising function.

**Reward.** We define the reward function to align with the measure $\mathcal{D}$, which is computed from the edited image $I^{edit}$ based on two editing criteria: background preservation and prompt alignment.

***Background preservation.*** The background region, defined as the area outside the editing mask $M$, should remain consistent with the original image $I^{src}$. We employ the mean squared error (MSE) to quantify this consistency since it is susceptible to subtle pixel-level changes. The background preservation reward is formulated as:

$$R_{noedit} = -\text{MSE}((1 - \mathbf{M}) \odot I^{src}, (1 - \mathbf{M}) \odot I^{edit}), \tag{1}$$

where $\odot$ denotes element-wise multiplication.

***Prompt Alignment.*** The content within the mask M of the edited image $I^{edit}$ should be semantically aligned with the edit prompt $p_{edit}$. We measure this alignment using the CLIP score [33], which computes the semantic similarity between the CLIP embeddings of the edited image and the text prompt. Denote $R_{edit}$ as the reward of the prompt alignment.

The total reward is computed as a weighted combination of the prompt alignment and background preservation rewards:

$$R(\mathcal{H}, I^{src}, p_{src}, p_{edit}) = \alpha R_{edit} + \beta R_{noedit}, \tag{2}$$

where $\alpha$ and $\beta$ are coefficients that govern the trade-off between edit effectiveness and background preservation. Empirically, we set $\alpha = \beta = 30$ as the default configuration to achieve an optimal equilibrium between the two objectives. This balanced weighting ensures robust alignment with the edit prompt while minimizing undesired alterations to the background. The sensitivity of model behavior to variations in $\alpha$ and $\beta$, along with further analysis of this trade-off, is detailed in Section 4.4.

***Global editing and small region editing*** For global editing tasks, such as style transfer, we set the mask to $M = 1$ across the entire image and define the reward as $R = R_{edit}$, following the evaluation protocol of the PieBench dataset. For small region editing, such as changing eye color, the CLIP score reward is less effective on small regions. To address this, we incorporate a Large Vision-Language Model (LVLM) to compute prompt alignment rewards. Specifically, for each edited image, given the original and edited prompts, we pose a question related to the editing instruction and let the model select the correct answer from multiple choices. The reward is defined as the total number of correct answers. We provide the ablation study and the effectiveness of using LVLM as the reward compared to the CLIP score reward in section 4.4.

**Termination.** The RL process terminates upon completing all $T$ denoising steps. We employ Proximal Policy Optimization (PPO) [37] for training, which comprises two core components: 1) a policy model $\pi_\theta(s_t) = \pi_\theta(x_t, t)$ that estimates action probabilities given states $s_t$, and 2) a value model $V_\tau(x_t, t)$ that predicts state rewards. Additional implementation details are provided in Section 3.2.3.

We frame hyperparameter optimization as a reinforcement learning problem where each denoising timestep corresponds to a unique state. This formulation induces an exponential state space growth ($N^T$ possible states), where $N$ represents the possible value of state. To address the challenge of this huge state space while leveraging the constrained range of the hyperparameters in image editing, we propose a two-phase RL strategy: **Phase 1 - Pretraining:** The policy is pretrained using hyperparameters randomly sampled from a predefined prior distribution. This constrains exploration to a promising subspace, significantly reducing state space complexity. The resulting model $\pi_{\theta_1}$ serves as initialization for subsequent learning. **Phase 2 - Online Learning:** Starting from $\pi_{\theta_1}$, the policy $\pi_{\theta_2}$ engages in environment interactions while continuously optimizing through

reward maximization. This stage enables optimal hyperparameter searching within the constrained hyperparameter space from **Phase 1**.

### 3.2.2 Phase 1 - Policy Initialization

In image editing, we exploit known priors for each hyperparameter, *e.g.*, inversion timesteps [15] commonly in $[20, 45]$ or attention-replacement ratios [12] commonly in $[0.2, 0.8]$. Let $p_0(\mathcal{H})$ denote these priors over $\mathcal{H} = \{\mathcal{H}_T, \dots, \mathcal{H}_1\}$. We train the policy model $\pi_\theta$ to align its outputs with $p_0$ by minimizing the objective function:

$$\theta_1 = \arg\min_\theta \ \mathbb{E}_{\mathcal{H} \sim p_0(\mathcal{H})} \left[ \frac{1}{T} \sum_{t=1}^{T} \mathcal{L}_1\big(\pi_\theta(x_t, t), \ \mathcal{H}_t\big) \right], \tag{3}$$

where $\mathcal{L}_1$ penalizes deviation from the sampled $\mathcal{H}_t$.

### 3.2.3 Phase 2 - Hyperparameters Identification

During online exploration, at each denoising step $t$, the policy samples $\mathcal{H}_t \sim \pi_\theta(x_t, t)$ (see Figure 2). This action reconstructs the previous latent $x_{t-1} = g(x_t, t, \mathcal{H}_t)$, advancing the denoising process. The full hyperparameter sequence $\mathcal{H} = \{\mathcal{H}_T, \dots, \mathcal{H}_1\}$ yields the edited image $I^{edit}$, from which we compute a reward $R$ in equation 2. We then update the policy by minimizing the following objective:

$$\mathcal{L}_2 = -\mathbb{E}_{\substack{\mathcal{H}_t \sim \pi_\theta(x_t, t) \\ (I^{src}, p_{src}, p_{edit}) \sim p_{data}}} \big[ R(\mathcal{H}, I^{src}, p_{src}, p_{edit}) - \eta \, D_{KL} \big], \tag{4}$$

where $p_{data}$ is the data distribution, and $D_{KL}$ regularizes the current policy $A_{\theta_2}$ against the stage-1 policy $\pi_{\theta_1}$ [30, 24, 20]. Since $\mathcal{H}_t \sim \pi_\theta(x_t, t)$, we only require $O(1)$ for each denoising step, resulting in $\mathcal{O}(T)$.

We also train the value model $V_\tau$ to predict the cumulative reward at each state by minimizing the error between its estimation and the actual return. In PPO fashion, the policy $\pi_\theta$ and value model $V_\tau$ are alternately optimized to maximize expected cumulative reward. (More information of training and testing phases could be found in the Appendix)

### 3.2.4 Network Design

We treat each pair $(x_t, t)$ as an RL state. We condition the policy on $x_t$, timestep $t$, and prompts $p_{src}, p_{edit}$ since the optimial hyperparameters depend on the provided prompts, which match with the U-Net input. A pretrained U-Net Encoder $U_\omega$ extracts two features $f_{src} = U_\omega(x_t, t, p_{src})$, $\quad f_{edit} = U_\omega(x_t, t, p_{edit})$. These feature maps are concatenated, then passed through a $2 \times 2$ convolution and spatial average pooling layer to yield $f_{\text{cat}}$. Timestep $t$ is encoded sinusoidally and linearly projected to $f_t$. We concatenate $[f_{\text{cat}}; f_t]$, process it with two ReLU-activated fully connected layers, then use $K$ separate linear heads to predict $K$ hyperparameters. The value model shares this backbone architecture but output a single scalar output. Only the newly added layers are trained (see Fig. 2).

## 4 Experiment

### 4.1 Dataset Setup

Our training data originates from the EditBench collection [22], with task selection strictly aligned to PieBench [16] benchmark objectives. To establish a coherent framework for prompt-to-prompt editing, each training datapoint requires four elements: the original image, the original prompt (generated by ChatGPT based on image content), the edit prompt (derived through ChatGPT analysis of edited images and instruction prompts), and the edit mask (object segmentation via SAM [19]). This curation process involves three systematic steps: **Object Localization**: ChatGPT extracts target editing objects from instruction prompts through cross-modal analysis. **Prompt Engineering**: Dual prompts (original/edited) are synthesized by ChatGPT using image-object contextual relationships. **Mask Generation**: SAM precisely segments identified objects from original images. The resulting dataset contains 2,000 rigorously annotated samples. For evaluation, we adopt the PieBench dataset [16], which contains 700 samples across diverse editing scenarios, ensuring comprehensive capability assessment.

## 4.2 Implementation Details

We choose denoising step $T = 50$ by default. During **Phase-1** training, we implement parameter initialization policies based on different editing method requirements. 1) For methods requiring inversion timestep searching, we randomly sample the inversion timestep within the range of $0.35 - 0.95$ of the total denoising timesteps $T$; 2) For methods involving cross-attention or self-attention replacement ratios, we initialize these parameters by uniform sampling from their default range of $0.2 - 0.8$; 3) For scalar hyperparameters (*e.g.*, attention weights in [12]), we instruct the policy model to predict their default values. More details of the prior for each type of hyperparameter can be found in the Appendix. In **Phase-2**, we integrate the **Phase-1** model into the training framework to compute the KL divergence term $D_{KL}$ in Equation 4, following the implementation strategies in [24, 30]. Both policy and value models are optimized using Adam [18] with a fixed learning rate of $5 \times 10^{-5}$. For the reward function configuration, we set the coefficients $\alpha = 30$ and $\beta = 30$ as default values unless otherwise specified. Consistent with prior works [24, 30], the KL divergence coefficient remains $\gamma = 0.02$ across all experiments.

## 4.3 AutoEdit Improves the Performance over Other Methods

**Baselines.** We evaluate AutoEdit across multiple image editing frameworks, including training-free methods such as DDIM-Inversion [39], DDPM-Inversion [15], PnP-Inversion [16], Prompt-to-Prompt [12], MasaCtrl [5], and Null-text Inversion [25], as well as training-based methods such as InstructPix2Pix [4] and UltraEdit [47]. Our experiments span a variety of base models, including SD 1.4, SD 1.5, SDXL, and DiT. The hyperparameter settings for each method such as inversion timestep ranges and attention replacement ratios are systematically reported in the Appendix.

**Metrics.** Following the PieBench benchmark [16], our evaluation protocol measures four key aspects of editing performance: (i) structural consistency, quantified by Structure Distance (SD); (ii) background preservation, assessed with classical metrics including PSNR, SSIM, MSE, and LPIPS; and (iii) semantic alignment, evaluated via CLIP scores (ViT-B/32) computed separately for edited regions and full images. To further validate the quality of edits, particularly on small regions, we additionally incorporate an evaluation based on the judgment of a Large Vision-Language Model (LVLM), referred to as the LLM Score. For each image and its corresponding editing instruction, the LVLM is prompted with a question derived from the instruction, and its response is used to quantify editing quality. Details of question construction and the LLM Score evaluation protocol are provided in the Appendix.

**Quantitative Results.** We present the main experimental results in Table 1. We report the best performance for the baseline methods with the hyperparameter configuration that yields the highest reward. For DDIM Inversion [39] and DDPM Inversion [15], the best configuration of the inversion timestep $r$ is $35$ and $40$, respectively. Under these configurations, the models achieve high CLIP scores but struggle with maintaining background consistency. When integrated with AutoEdit, both methods demonstrate significantly improved background preservation, accompanied by only a slight drop in CLIP scores. We contend that this minor decrease has minimal impact on editing fidelity due to the inherent insensitivity of the CLIP metric. Moreover, these methods inherently face a trade-off between editing fidelity and background preservation. Our proposed reward addresses this issue by offering a more balanced assessment of editing performance. Additional visual evidence of this trade-off is provided in the Appendix. Furthermore, AutoEdit consistently improves performance on the LLM Score metric for both inversion-based methods [39, 15]. For P2P [12], PnP [16], and MasaCtrl [5], AutoEdit leads to enhanced performance in both background preservation and CLIP editing scores, underscoring its effectiveness in guiding hyperparameter identification.

We further evaluate training-based approaches, including InstructPix2Pix [4] and UltraEdit [47], by applying AutoEdit to adaptively determine the CFG coefficient during the denoising process. Compared to the default fixed value of 7.5, the adaptive CFG strategy improves background preservation metrics while maintaining prompt alignment performance.

Our experiments span a diverse set of editing methods and base models, demonstrating the generalization of AutoEdit. More results of AutoEdit on flow-based editing method is in Table 6.

**Qualitative Results.** We present qualitative results in Figure 3. For each baseline, we visualize the output generated using its default hyperparameters, as specified in Table 1. Overall, AutoEdit consistently improves the performance of all editing methods by selecting more suitable hyperparameter

Table 1: **The comparison with popular image editing methods**. Note that AutoEdit enables improvement over baselines with only negligible cost.

| Method | Base Model | Structure Distance ↓ | Background Preservation | | | | CLIP Score | | LLM Score |
|---|---|---|---|---|---|---|---|---|---|
| | | | PSNR ↑ | SSIM ↑ | MSE ↓ | LPIPS ↓ | Edited ↑ | Whole ↑ | Score |
| DDIM-Inversion [39] | SD 1.4 | 38.10 | 21.36 | 76.67 | 103.95 | 146.60 | **23.30** | **26.31** | 0.96 |
| + AutoEdit | | **18.74** | **24.65** | **81.28** | **52.94** | **95.10** | 22.65 | 25.72 | **1.12** |
| DDPM-Inversion [15] | SD 1.4 | 22.12 | 22.66 | 78.95 | 53.33 | 67.66 | **23.02** | **26.22** | 1.03 |
| + AutoEdit | | **12.65** | **27.25** | **85.17** | **31.18** | **50.51** | 22.52 | 25.83 | **1.17** |
| PnP Inversion [16] | SD 1.5 | 11.65 | 27.22 | 84.76 | 35.86 | 60.67 | 22.10 | 25.02 | 1.10 |
| + AutoEdit | | **11.06** | **27.85** | **85.04** | **33.77** | **60.12** | **23.00** | **25.79** | **1.19** |
| P2P [12] | SD 1.4 | 14.75 | 25.82 | 84.02 | 40.93 | 61.78 | 22.29 | 25.44 | 1.08 |
| + AutoEdit | | **13.76** | **26.45** | **84.08** | **36.24** | **60.60** | **23.88** | **26.55** | **1.22** |
| MasaCtrl [5] | SD 1.4 | 28.38 | 22.17 | 79.67 | 86.97 | 79.67 | 21.16 | 23.96 | 0.92 |
| + AutoEdit | | **21.33** | **23.48** | **80.06** | **46.28** | **71.35** | **21.75** | **24.86** | **0.99** |
| DDPM-Inversion [15] | SDXL | 7.12 | 26.13 | 89.88 | 35.32 | 65.62 | **23.0** | **27.11** | 1.19 |
| +AutoEdit | | **6.46** | **27.86** | **90.50** | **20.44** | **53.51** | 22.9 | 26.7 | **1.27** |
| UltraEdit [47] | MM-DiT | 10.82 | 26.5 | 84.7 | 46.7 | 75.8 | 22.4 | 25.6 | 1.20 |
| +AutoEdit | | **7.61** | **27.3** | **86.2** | **37.6** | **64.9** | **22.6** | **25.7** | **1.26** |
| InstructPix2Pix [4] | SD 1.5 | 35.37 | 20.8 | 76.4 | 226.8 | 157.3 | 22.1 | 24.5 | 0.65 |
| +AutoEdit | | **28.68** | **22.2** | **78.5** | **181.4** | **132.8** | **22.3** | **24.7** | **0.82** |
| Null-text [25] | SD 1.4 | 19.87 | 23.8 | 79.9 | 64.4 | 109.8 | 22.3 | 25.9 | 1.12 |
| +AutoEdit | | **10.91** | **25.7** | **82.4** | **45.4** | **82.3** | **22.6** | **26.3** | **1.21** |

configurations. Our qualitative analysis highlights several common failure cases resulting from sub-optimal hyperparameter choices: inaccurate background reconstruction (*e.g.*, the stone background in the cat image for DDIM-Inversion and DDPM-Inversion), unnatural object synthesis (*e.g.*, the distorted rock in P2P), failure to execute the desired edit (*e.g.*, the unaltered stone in MasaCtrl or the unchanged noodles in MasaCtrl, PnP, and DDIM-Inversion), and noticeable facial discrepancies (*e.g.*, the altered woman's face in DDPM-Inversion and P2P). These findings demonstrate that improved hyperparameter selection, facilitated by AutoEdit, can substantially enhance editing quality, even for weak baseline methods such as DDIM-Inversion.

**Policy Model Behavior in Inference.** We analyze the behavior of key hyperparameters predicted by the policy model during inference. For the inversion timestep $t$ in DDPM-Inversion [15], where the policy chooses between $p_{src}$ and $p_{edit}$, we define the inversion timestep as the first step $t$ where the policy selects $p_{edit}$. Figure 4a shows the distribution of inversion timesteps chosen by the policy model. We observe that the policy most frequently selects $t$ between 25 and 40, which aligns with common choices in prior editing methods. A similar analysis is conducted for the cross-attention replacement ratio $r$ in P2P [12], with results shown in Figure 4b. We find that $r$ is most often selected between 0.3 and 0.6. Importantly, both $t$ and $r$ vary significantly across samples, indicating that AutoEdit adapts its hyperparameter predictions for each specific image to optimize the final reward.

**Reward Across Training Episodes.** We plot the reward of AutoEdit on DDPM-Inversion [15] over the first 3000 training episodes. The results are shown in Figure 4c, where episode 0 corresponds to the **Phase-1** model $A_{\theta_1}$. We observed a consistent increase in reward during PPO training up to episode 2500, after which it slightly declined at episode 3000. Based on this trend, we limit training to the first 2500 episodes (approximately 15 epochs).

### 4.4 Ablation Study

**Effect of Reward Coefficients of $\alpha$ and $\beta$.** We investigate the impact of different values of $\alpha$ and $\beta$ in the reward function. Using

Table 2: The necessity of **Phase-1** (policy initialization).

| P1 | P2 | PSNR ↑ | SSIM ↑ | MSE ↓ | LPIPS ↓ | Edited ↑ | Whole ↑ | Reward |
|---|---|---|---|---|---|---|---|---|
| | ✓ | 18.2 | 74.5 | 208.7 | 57.9 | **23.2** | **26.3** | 6.12 |
| ✓ | | 22.1 | 77.4 | 52.7 | 69.7 | 20.7 | 23.4 | 5.42 |
| ✓ | ✓ | **27.2** | **85.3** | **31.1** | **50.5** | 22.5 | 25.8 | **6.25** |

DDPM-Inversion [15] as the baseline, we fix $\alpha = 30$ and train AutoEdit with varying values of $\beta$. All models are initialized from the same **Phase-1** checkpoint $A_1$. Table 3 summarizes the behavior of AutoEdit under different $\beta$ settings. When $\beta$ is small (*e.g.*, $\beta = 10$ or $\beta = 20$), the reward function

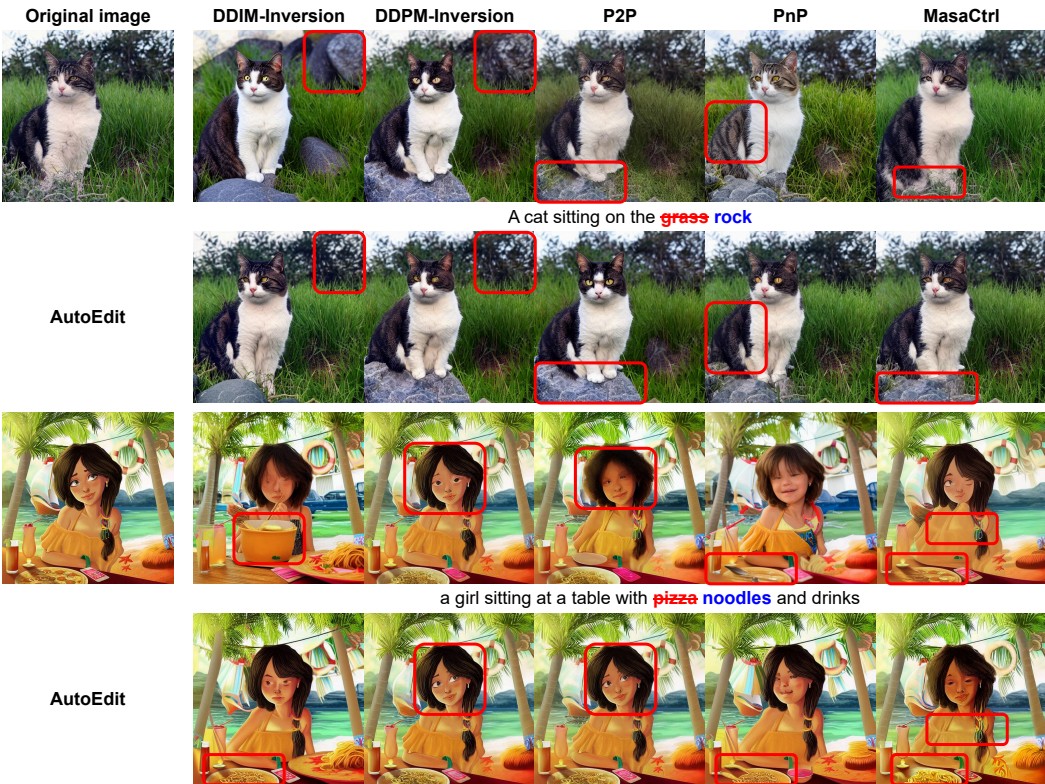

A cat sitting on the ~~grass~~ **rock**

a girl sitting at a table with ~~pizza~~ **noodles** and drinks

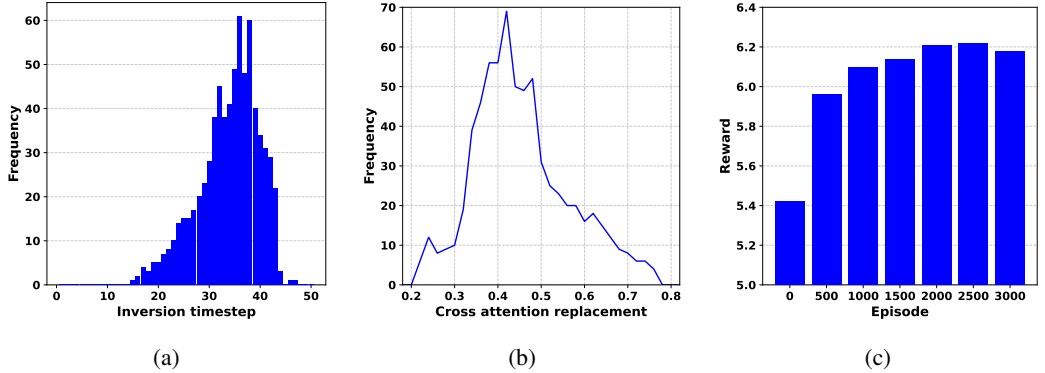

Figure 3: We compare the qualitative results of AutoEdit with the default hyperparameter choice of the baseline. Overall, AutoEdit can search for better hyperparameters, resulting in better object editing, background preservation, and more natural images.

Figure 4: The analysis of a) inversion timestep, b) cross-attention replacement, and c) the reward during the training of the policy model $A_\theta$.

emphasizes the CLIP score more. As a result, AutoEdit achieves higher CLIP scores, but at the cost of reduced background preservation. Conversely, when $\beta$ is increased to 40, the model prioritizes background preservation, as it becomes more influential in the reward signal. $\beta = 30$ strikes a good balance between CLIP alignment and background consistency. Figure 5 provides qualitative examples illustrating the effects of different $\beta$ values.

**Different choice of the reward.** We find that the CLIP score performs poorly on small region image editing tasks with a limited edit mask $M$, such as eye color modification. As a result, using CLIP as the reward fails to reliably capture editing quality. In contrast, Large Vision-Language Models (LVLMs) can effectively assess localized edits, for example, verify whether the eye color has been correctly changed, as highlighted by recent editing benchmarks [22, 44]. Since our proposed RL framework is generic, it can incorporate any reward function that reflects editing quality. Accordingly, we replace the CLIP-based reward with an LVLM-based evaluation. Our LVLM reward is similar to

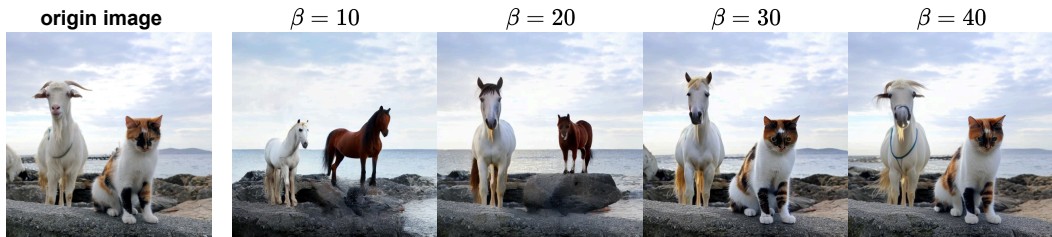

photo of a ~~goat~~ **horse** and a cat standing on rocks near the ocean

Figure 5: Edited image with different value of $\beta$

Table 3: Comparison of the behavior of AutoEdit with the different choices of $\alpha$ and $\beta$. When we decrease the $\beta$, AutoEdit optimizes the CLIP score. Otherwise, it tries to preserve the background.

| $\alpha, \beta$ | PSNR ↑ | SSIM ↑ | MSE ↓ | LPIPS ↓ | Edited ↑ | Whole ↑ |
|---|---|---|---|---|---|---|
| $\alpha = 30, \beta = 10$ | 19.65 | 77.11 | 150.5 | 138.6 | 24.15 | 27.34 |
| $\alpha = 30, \beta = 20$ | 23.59 | 82.15 | 66.84 | 82.30 | 23.44 | 26.95 |
| $\alpha = 30, \beta = 30$ | 27.25 | 85.17 | 31.18 | 50.51 | 22.52 | 25.83 |
| $\alpha = 30, \beta = 40$ | 28.53 | 86.03 | 24.72 | 42.80 | 21.36 | 24.36 |

the LLM Score metric, which is described in detail in the Appendix. Table 4 reports the performance of AutoEdit under LVLM evaluation. We observe consistent improvements in both background preservation metrics and LLM Score, while maintaining strong prompt alignment performance.

**Policy Initialization.** We conduct experiments to highlight the importance of **Phase-1** training (policy initialization). Using DDPM-Inversion [15] as the baseline, we compare results with

Table 4: Comparison of AutoEdit performance with LLM Score as the reward function

| Method | PSNR | SSIM | MSE | LPIPS | Edited | Whole | LLM |
|---|---|---|---|---|---|---|---|
| DDPM Inv | 26.1 | 89.8 | 35.3 | 65.6 | 23.0 | 27.1 | 1.19 |
| + AutoEdit | 27.8 | 90.5 | 20.4 | 53.5 | 22.9 | 26.7 | 1.27 |
| + AutoEdit + LLM | **29.1** | **91.8** | **19.1** | **49.1** | 22.7 | 26.6 | **1.31** |

and without **Phase-1**. The outcomes are summarized in Table 2. As shown in the table, omitting **Phase-1** causes the model to apply overly strong edits while failing to preserve the background, resulting in a substantially lower background preservation score. We attribute this behavior to the difficulty of exploring the large state space introduced by the 50-step denoising process in DDPM. In contrast, incorporating **Phase-1** improves background preservation and CLIP alignment, demonstrating the effectiveness of this initialization phase in guiding policy learning.

**Computational Cost.** We compare the inference time and number of parameters between the original U-Net and AutoEdit for a single image-prompt pair in DDPM-Inversion. Table 5 reports the inference time of AutoEdit on Stable Diffusion. Our method introduces only negligible overhead in inference time and model size compared to the original U-Net.

Table 5: Computational cost of AutoEdit is negligible.

| Method | Inference (ms) | #Params |
|---|---|---|
| SD Unet | 20.89 | 86M |
| +AutoEdit | 21.03 (+0.6%) | 87.1M (+0.6%) |

## 5 Conclusion

We propose AutoEdit, an RL-based framework for per-image hyperparameter identification in diffusion-model editing. By casting the denoising process as a reinforcement learning problem, AutoEdit searches within a single trajectory, eliminating the exponential overhead of trial-and-error. Empirical results across multiple editing methods demonstrate that AutoEdit improves editing quality while maintaining runtime efficiency.

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

# A  AutoEdit baselines

In this section, we provide the details of the hyperparameters used in each baseline method:

**Baselines**

***DDIM/DDPM Inversion*** [39, 15]: These methods first invert the image to a latent representation $x_r$ using the source prompt $p_{src}$, followed by a denoising process from timestep $r$ to 0 conditioned on the edit prompt $p_{edit}$. The only required hyperparameter in this process is the inversion timestep, i.e., $\mathcal{H} = \{r\}$. Due to the reversibility of the DDIM/DDPM framework, each denoising step can be viewed as an action $\mathcal{H}_t = \{h_t\}$, where $h_t \in \{p_{src}, p_{edit}\}$. By default, the inversion timestep is set to $r = 35$ for DDIM-Inversion and $r = 40$ for DDPM-Inversion.

***P2P*** [12]: introduces three key hyperparameters: the inversion timestep $r$, the cross-attention replacement ratio $u$, and the attention weight $w$, summarized as $\mathcal{H} = \{r, u, w\}$. The inversion timestep $r$ can be handled similarly to the strategy used in DDIM/DDPM-Inversion. The cross-attention ratio $u$ modifies the attention mechanism in the U-Net by replacing the cross-attention conditioned on $p_{\text{edit}}$ with that of $p_{\text{src}}$ during the first $u \cdot T$ denoising steps, where $T$ is the total number of steps. This decision is modeled as a binary action at each timestep: whether to replace the cross-attention or not. The attention weight $w$ scales the cross-attention values for the edited word, determining the emphasis placed on that word in the output image. We discretize $w$ by allowing the policy to select from a fixed set of values $\{0.5, 1.0, 1.5, 2.0, 3.0, 5.0\}$. Formally, we define the action space at each timestep as $\mathcal{H}_t = \{h_t^1, h_t^2, h_t^3\}$, where $h_t^1 \in \{p_{src}, p_{edit}\}$, $h_t^2 \in \{0, 1\}$ (with 1 indicating replacement), and $h_t^3 \in \{0.5, 1.0, 1.5, 2.0, 3.0, 5.0\}$. By default, the inversion timestep is set to $r = 10$, the cross-attention ratio to $u = 0.4$, and the attention weight to $w = 1.0$.

***PnP Inversion*** [16]: In this method, two binary hyperparameters are defined at each denoising step, denoted as $\mathcal{H}_t = \{h_t^1, h_t^2\}$. The first, $h_t^1 \in \{0, 1\}$, determines whether to replace the self-attention computation in the edit branch with that from the unconditional branch. The second, $h_t^2 \in \{0, 1\}$, controls whether to replace the convolutional features of the edit branch with those from the unconditional branch. In the default setting of PnP Inversion, self-attention is replaced during the first 80% of the denoising process, while convolutional features are replaced during the first 50% of the steps.

***MasaCtrl*** [5]: At each denoising step, the method decides whether to replace the self-attention in the edit branch with that from the unconditional branch. This decision is represented by a binary hyperparameter $\mathcal{H}_t = \{h_t\}$, where $h_t \in \{0, 1\}$. By default, the replacement is applied starting from timestep $t = 4$ of the denoising process.

***UltraEdit*** [47] and ***InstructPix2Pix*** [4] Both methods are training-based editing method. Our RL framework is used to optimize the CFG coefficient during the sampling process of both methods. The default CFG coefficient is set to 7.5.

***Null-text inversion*** [25] Null-text is an inversion method that addresses the mismatch between inversion and denoising process through CFG by optimizing the null embedding. In this method, we use P2P as the editing operation. We only use RL to optimize the inversion timestep and cross attention ratio $u$. Similar to P2P, we set the default value of $r = 40$ and $u = 0.4$.

**Phase-1 prior for each type of hyperparameters**: The inversion timestep $r$ is randomly sampled from the range $[50 - 0.65T, 50 - 0.05T]$. The cross-attention ratio is similarly sampled from the range $[0.2, 0.6]$. The prior for the attention weight $w$ is fixed at 1.0. For PnP Inversion [16], the ratios for replacing self-attention and convolutional features are randomly sampled from the range $[0.2T, 0.8T]$. Lastly, in MasaCtrl [5], self-attention replacement begins at timestep $t$, where $t$ is randomly selected between step 4 and step 20.

**Phase-1 loss**: Since all the parameterizations of $\mathcal{H}_t$ are discrete, we choose the loss $\mathcal{L}_1$ for **Phase-1** training to be the cross-entropy loss. After sampling $\mathcal{H} \sim p_0(\mathcal{H})$, the loss $\mathcal{L}_1$ is applied at each timestep to enforce the policy model's prediction to match $\mathcal{H}_t$, the timestep-specific parameterization of $\mathcal{H}$ at $t$.

Table 6: Results of AutoEdit on a Flow-based image editing method

| Method | PSNR | SSIM | CLIP Edit | CLIP Whole | LLM Score |
|---|---|---|---|---|---|
| Taming flow [42] | 23.4 | 81.5 | 22.9 | 26.0 | 1.22 |
| +AutoEdit | **25.7** | **85.2** | **23.4** | **26.1** | **1.30** |
| Fireflow [6] | 23.1 | 82.2 | 22.4 | 25.2 | 1.20 |
| +AutoEdit | **26.2** | **86.2** | **22.9** | **25.2** | **1.27** |

## B More results on flow-based editing

We present the results of AutoEdit applied to flow-based editing methods, as shown in Table 6. Specifically, AutoEdit is used to search for the optimal injection timestep, a key hyperparameter in both Taming Flow [42] and Fireflow [6]. Overall, AutoEdit enhances the baseline performance in terms of both background preservation and prompt alignment, demonstrating its effectiveness in automatically identifying hyperparameters that improve editing results.

## C Global editing

For global editing tasks, such as style transfer, the editing mask is set to $M = 1$ for the entire image. In this case, $R_{noedit} = 0$, and the reward function $R$ reduces to the CLIP score. To assess the effectiveness of AutoEdit in this setting, we report CLIP scores in Table 7, comparing against the baseline methods DDPM-Inversion [15] and P2P [12]. The results show that AutoEdit consistently improves editing performance on the style transfer task relative to the baselines.

Table 7: AutoEdit on global editing task

| Base model | Editing method | CLIP score |
|---|---|---|
| SDXL | DDPM-Inv [15] | 26.5 |
| SDXL | +AutoEdit | **28.3** |
| SD 1.4 | P2P | 25.5 |
| SD 1.4 | +AutoEdit | **26.7** |

## D Convergence of PPO.

We evaluate how closely AutoEdit approaches the optimal reward through PPO training. In this experiment, we compare the reward achieved by AutoEdit against the best possible reward obtained from combinations of $k$ different hyperparameter values, where $k \in \{1, 2, 3\}$. We compute the reward for each image using $k$ different hyperparameter settings and select the maximum among them. The $k$ values are selected from the set that yields the highest possible reward.

Table 8: Comparison of AutoEdit 's reward with baselines using multiple optimal hyperparameter sets. AutoEdit achieves performance comparable to the best of three hyperparameters, demonstrating its convergence to a near-optimal reward.

| Method | #Trials | | | AutoEdit | Optimal |
|---|---|---|---|---|---|
| | 1 | 2 | 3 | | |
| DDIM-Inversion | 5.81 | 6.03 | 6.11 | 6.09 | 6.17 |
| DDPM-Inversion | 6.11 | 6.21 | 6.23 | 6.25 | 6.32 |
| P2P | 6.17 | 6.31 | 6.37 | 6.38 | 6.45 |
| MasaCtrl | 5.47 | 5.59 | 5.65 | 5.65 | 5.75 |

## E LLM Score

For small-region editing, the CLIP score fails to reliably capture editing quality. As illustrated in Figure 6, when the objective is to change a dog's eye color to blue, computing the CLIP score on a small mask $M$ with the prompt "blue eye" yields a lower score for the correctly edited image. This highlights the inaccuracy of CLIP-based rewards in such scenarios.

Previous evaluation benchmarks [22, 44] have adopted LLM judgment as an evaluation tool. A key advantage of LLM-based evaluation is its ability to verify whether small region edits are correct, as illustrated in Figure 6. Since our RL framework is generic, we replace the CLIP based reward with an LLM-based evaluation, which we denote as LLMScore. This score not only measures the quality

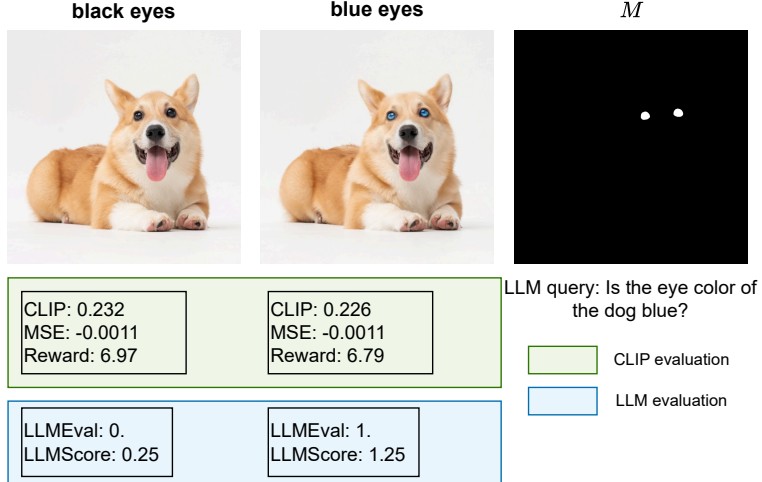

Figure 6: The advantage of LLM evaluation compared with CLIP score in small editing region

of an edited image but also serves directly as a reward function. More specifically, similar to the original reward, LLMScore consists of two components:

- **Prompt alignment** For each editing instruction, we construct a binary ("Yes/No") question to verify whether the required edit is present in the generated image. The edited image is then provided to the LLM, which outputs a response. We assign $R_{edit} = 1$ if the response is correct and $R_{edit} = 0$ otherwise.
- **Background preservation** For background preservation, we adopt MSE as the primary metric, as it is most sensitive to small changes in the background region. Accordingly, we define the background preservation reward as: $R_{noedit} = 1 - \gamma MSE((1-M) \odot I_{edit}, (1-M) \odot I_{src})$

The LLM score is computed as LLM Score $= R_{edit} + R_{noedit}$. We found $\alpha = 5$ works best in the editing task. For the LLM model, we adopt QwenVL-2.5-7B [2] as the LLM model.

## F  Trade-off between edit alignment and background preservation in DDIM/DDPM Inversion

In DDIM and DDPM Inversion [39, 15], a trade-off arises between alignment with the edit prompt and preservation of the original background. A high CLIP score typically indicates strong alignment with the edit prompt, but often at the cost of background fidelity, and vice versa. This trade-off is illustrated in Figure 7. Our proposed reward function offers a meaningful evaluation of edited images by explicitly balancing CLIP-based semantic alignment and background preservation (see Figure 7).

## G  Training and inference algorithm

We present the training and inference of AutoEdit in Algorithm 1, 2.

## H  Limitation

We illustrate the limitation of our approach in Figure 8. AutoEdit relies on the underlying editing method $\mathcal{E}$. Consequently, if $\mathcal{E}$ fails to perform a given edit, AutoEdit inherits this limitation. This issue is exemplified by the parrot case shown in Figure 8.

photo of a ~~goat~~ **horse** and a cat standing on rocks near the ocean

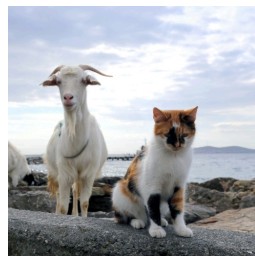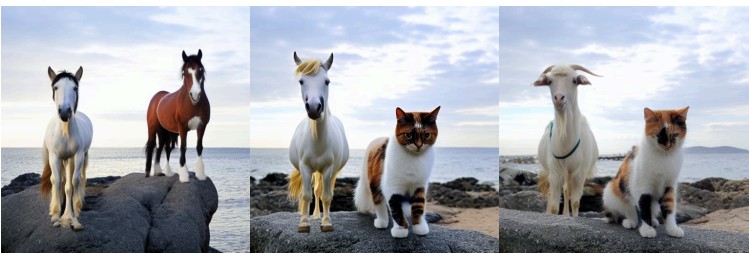

| | MSE: 0.111.
CLIP: 0.252
Reward: 4.23 | MSE: 0.024
CLIP: 0.207
Reward: 5.48 | MSE: 0.0054
CLIP: 0.174
Reward: 5.17 |

Figure 7: We provide the visualization of the trade-off between edit alignment and background preservation in DDIM-Inversion. Our reward can better reflect how good an edited image is.

---

**Algorithm 1:** AutoEdit Inference

---

**Input:** Editing method $\mathcal{E}$ with denoising function $g$ and inversion function $g^{-1}$, input image $I^{src}$, origin and edit prompts $p_{src}, p_{edit}$, pretrained policy model $\pi_\theta$, pretrained Stable Diffusion Model

**Output:** Edit image $I^{edit}$

**for** $t \leftarrow 1$ **to** $T$ **do**
  Perform diffusion inversion: $x_{t+1}^{src} = g^{-1}(x_t, t, p_{src})$.
**end**
$x_T^{edit} = x_T^{src}$
**for** $t \leftarrow T$ **to** $1$ **do**
  Sampling from the policy: $\mathcal{H}_t \sim \pi_\theta(x_t^{edit}, t)$.
  Denoise to get the previous noisy sample: $x_{t-1}^{edit} = g(x_t^{edit}, t, \mathcal{H}_t)$
**end**
VAE decode: $I^{edit} = VAE(x_0^{edit})$
**return** $I^{edit}$;

---

**Original image**    **DDPM-Inversion**    **AutoEdit**

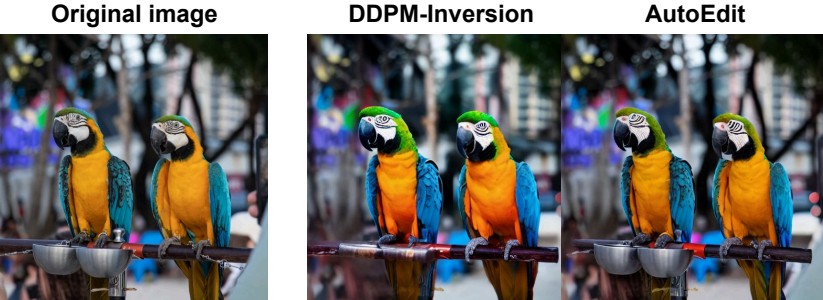

two **kissing** parrots sitting on a stick, city street background

Figure 8: The limitation of our approach. A key limitation of our approach is that if the underlying editing method performs poorly on a given image, AutoEdit is also likely to inherit and suffer from the same image.

**Algorithm 2:** PPO Training of AutoEdit

---

**Input:** Editing method $\mathcal{E}$ with denoising function $g$ and inversion function $g^{-1}$, policy model
$\pi_{\theta_2}$, **Phase 1** model $\pi_{\theta_1}$, pretrained Stable Diffusion Model, coefficient
$\beta = 0.02, \gamma = 0.999, \lambda = 0.95, \epsilon = 0.2$

**Inversion**:
$I^{src}, p_{src}, p_{edit} \sim p_{data}$
**for** $t \leftarrow 1$ **to** $T$ **do**
   | Perform diffusion inversion: $x_{t+1}^{src} = g^{-1}(x_t, t, p_{src})$.
**end**
**PPO exploration phase:**
**for** $t \leftarrow T$ **to** $1$ **do**
   | Get the distribution of action at current policy model: $\pi_t^2 = \pi_{\theta_2}(x_t^{edit}, t)$.
   | Get the distribution of action at **Phase-1** policy model: $\pi_t^1 = \pi_{\theta_1}(x_t^{edit}, t)$.
   | Sampling the hyperparameter: $\mathcal{H}_t \sim \pi_t^2$.
   | Compute the value: $v_t = V_\tau(x_t^{edit}, t)$.
   | Denoise to get the previous noisy latent: $x_{t-1}^{edit} = g(x_t^{edit}, t, \mathcal{H}_t)$
**end**
**Reward computation**:
Get the edit image $I^{edit} = VAE(x_0^{edit})$
Compute the reward $R = \alpha R_{edit}(I^{edit}, p_{edit}, M) + \beta R_{noedit}(I^{edit}, I^{src}, M)$
**Compute the return value and advantage recursively from the end of the denoising trajectory**:
$A_0 = 0$
**for** $t \leftarrow 1$ **to** $T$ **do**
   | $\delta = R + \gamma v_{t-1} - v_t$
   | Compute the advantage: $A_t = \delta + \gamma \lambda A_{t-1}$
   | Compute the return: $r_t = A_t + v_t$
**end**
**Training value model**: $\mathcal{L}_{vf} = \mathbb{E}_t[(V_\tau(x_t, t) - r_t)^2]$
**Training the policy model**:
Compute ratio $u_t = \exp(\log \pi_{\theta_2}(x_t, t) - \log(\pi_t^2))$.
Clip objective: $\mathcal{L}_{clip} = \min(A_t u_t, A_t clip(u_t, 1 - \epsilon, 1 + \epsilon))$
Policy loss: $\mathcal{L}_{pg} = -\mathbb{E}_t[\mathcal{L}_{clip} - \beta D_{KL}(\pi_t^2 \| \pi_t^1)]$
Update $\theta_2, \tau$ based on gradient optimization.
**return** $\pi_{\theta_2}$

---

# I  User study

We conducted a user study involving 50 participants to evaluate the performance of AutoEdit. Each survey question presented a side-by-side comparison between AutoEdit and the same editing method with two different sets of hyperparameters. The survey comprised a total of 30 questions. Overall, 82.78% of the responses favored the results produced by AutoEdit over the baseline, indicating a clear human preference for the outputs generated by our method.

# J  Additional qualitative examples

We present additional qualitative examples comparing the editing baselines with and without AutoEdit in Figures 9, 10, 11, 12. Furthermore, we compare AutoEdit against the same methods using different hyperparameter settings in Figures 14 and 15. Overall, AutoEdit achieves consistently strong editing results across various hyperparameter configurations, thereby significantly reducing the manual tuning effort required by practitioners. We also include the qualitative results of AutoEdit applying to the Flux-based editing [42, 6] methods, as shown in Figure 13

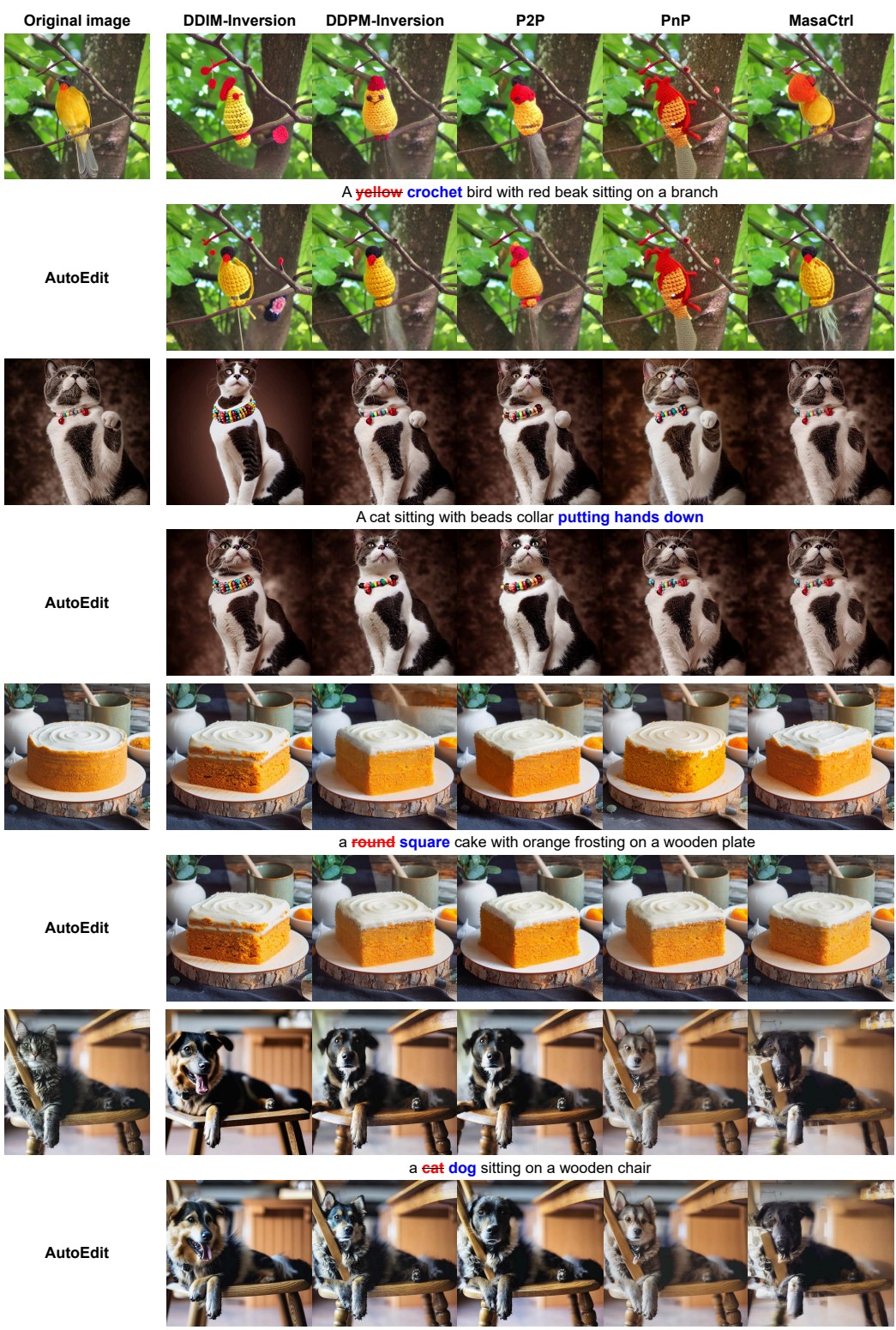

Figure 9: Additional qualitative samples

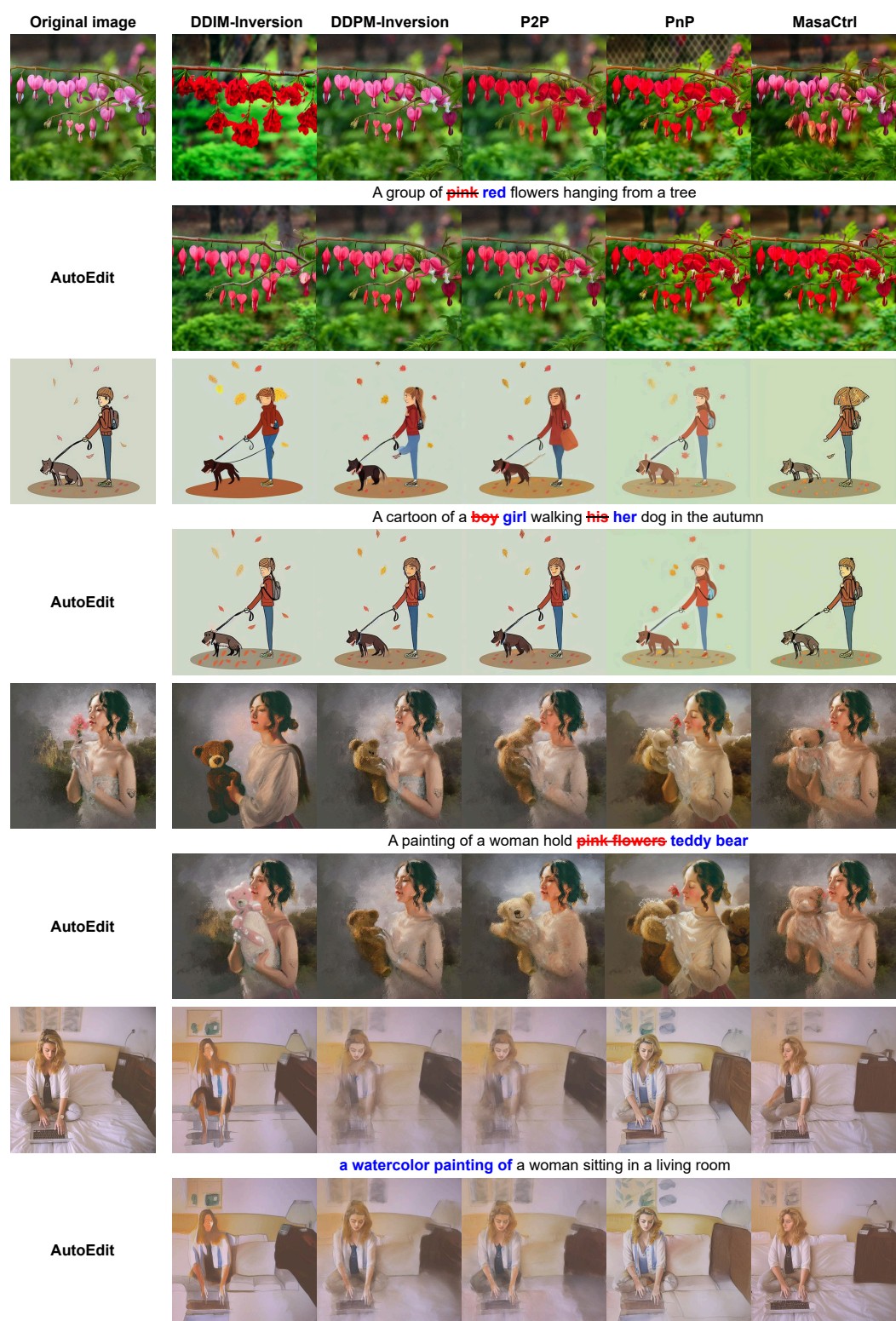

Figure 10: Additional qualitative results

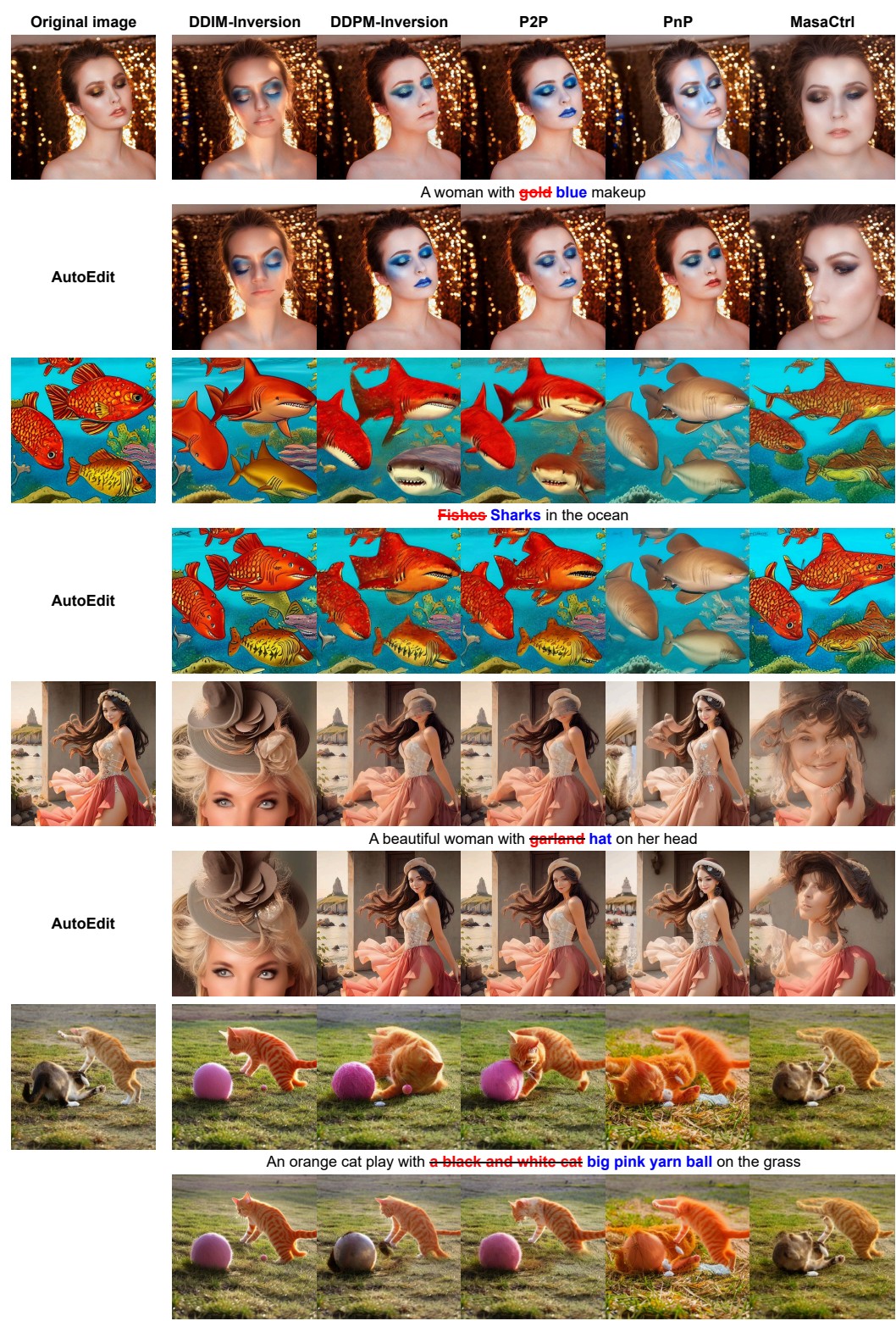

Figure 11: Additional qualitative examples

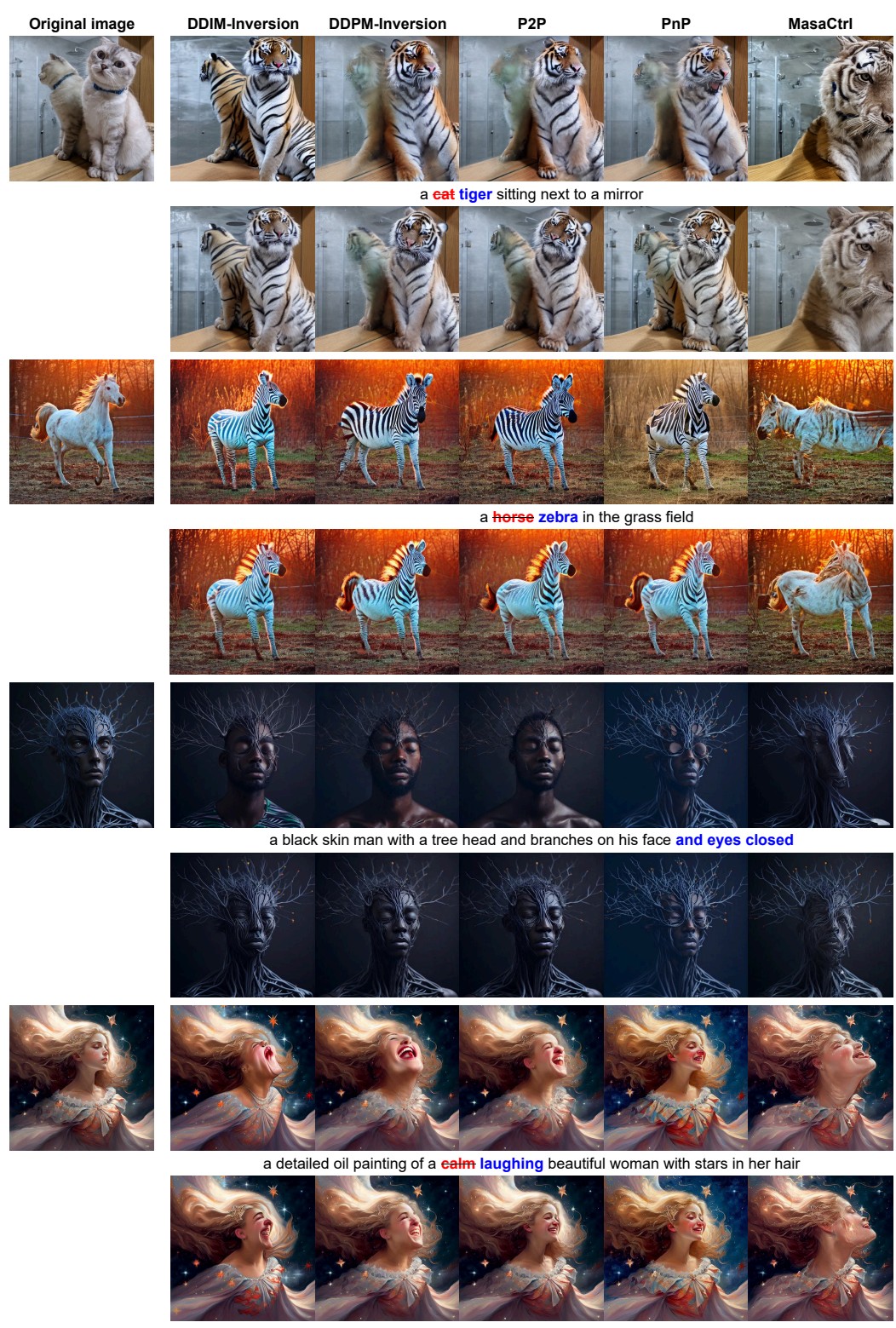

Figure 12: Additional qualitative examples

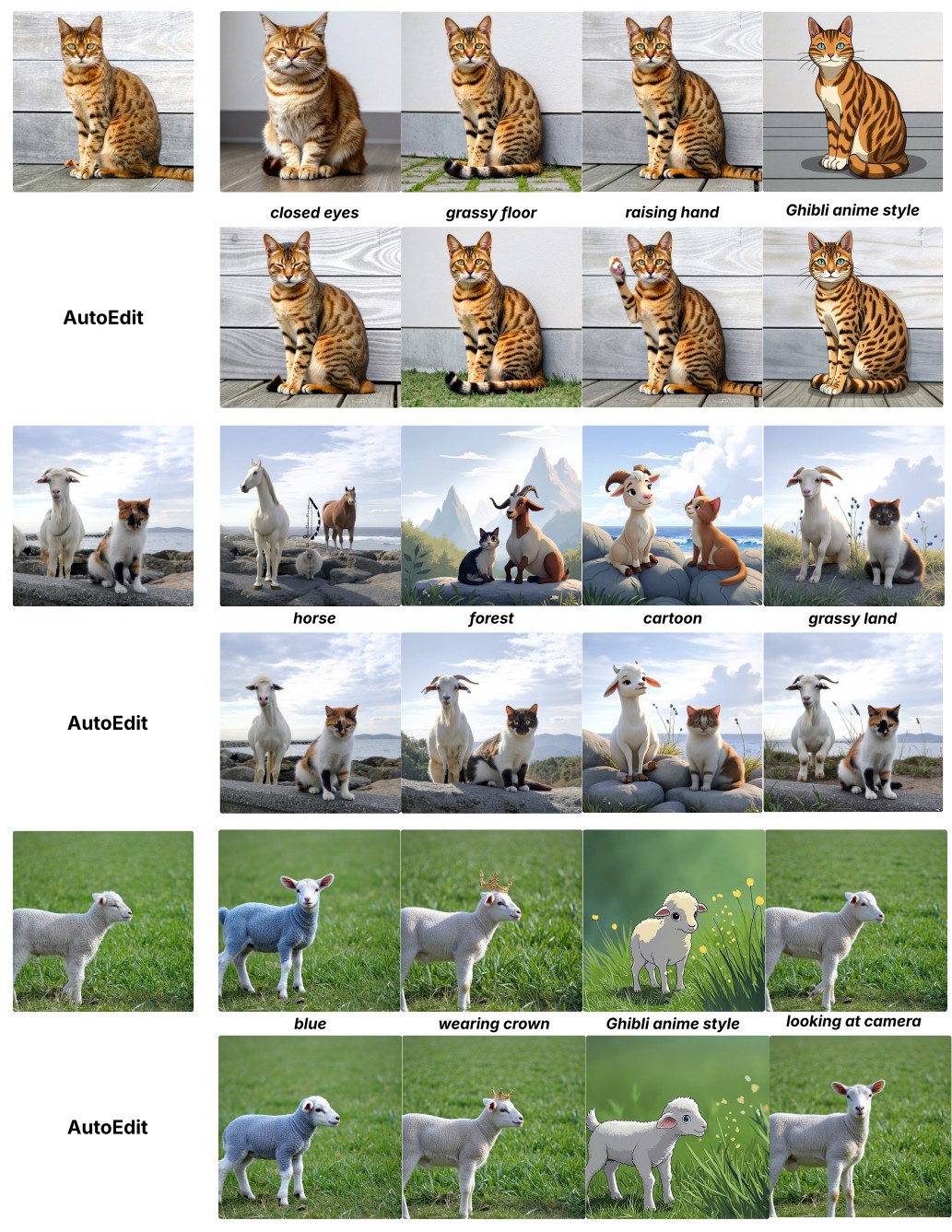

Figure 13: Qualitative Results on the Flux-based editing. **Top row**: The baseline with the default hyperparameter choice. **Bottom row**: AutoEdit performance. **The prompt** indicates the modification to the original image. This demonstrates the AutoEdit capability to reduce the running required to choose a good hyperparameter.

| Original image | 10 | 20 | 30 | AutoEdit |
|---|---|---|---|---|

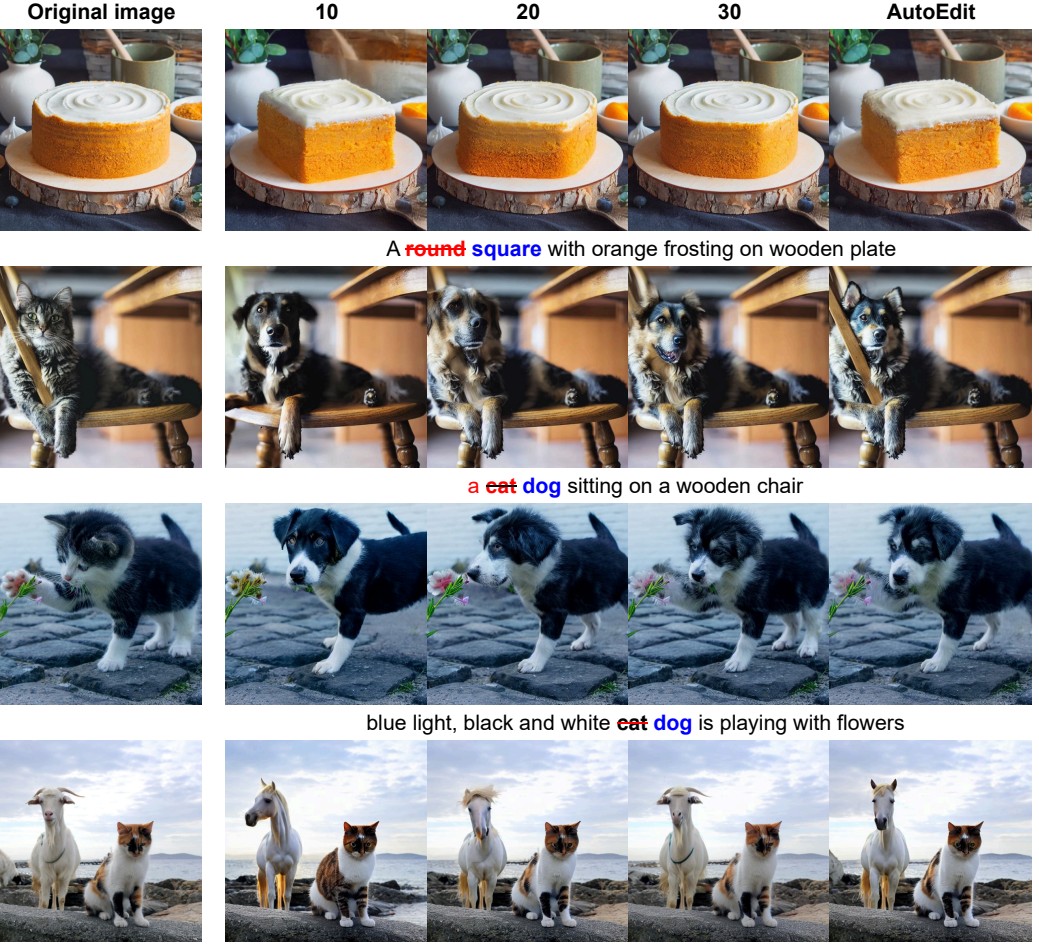

A ~~round~~ **square** with orange frosting on wooden plate

a ~~cat~~ **dog** sitting on a wooden chair

blue light, black and white ~~cat~~ **dog** is playing with flowers

Figure 14: Compare DDPM-Inversion with different values of inversion timestep. In general, AutoEdit can achieve on par or better performance without tuning the inversion timestep

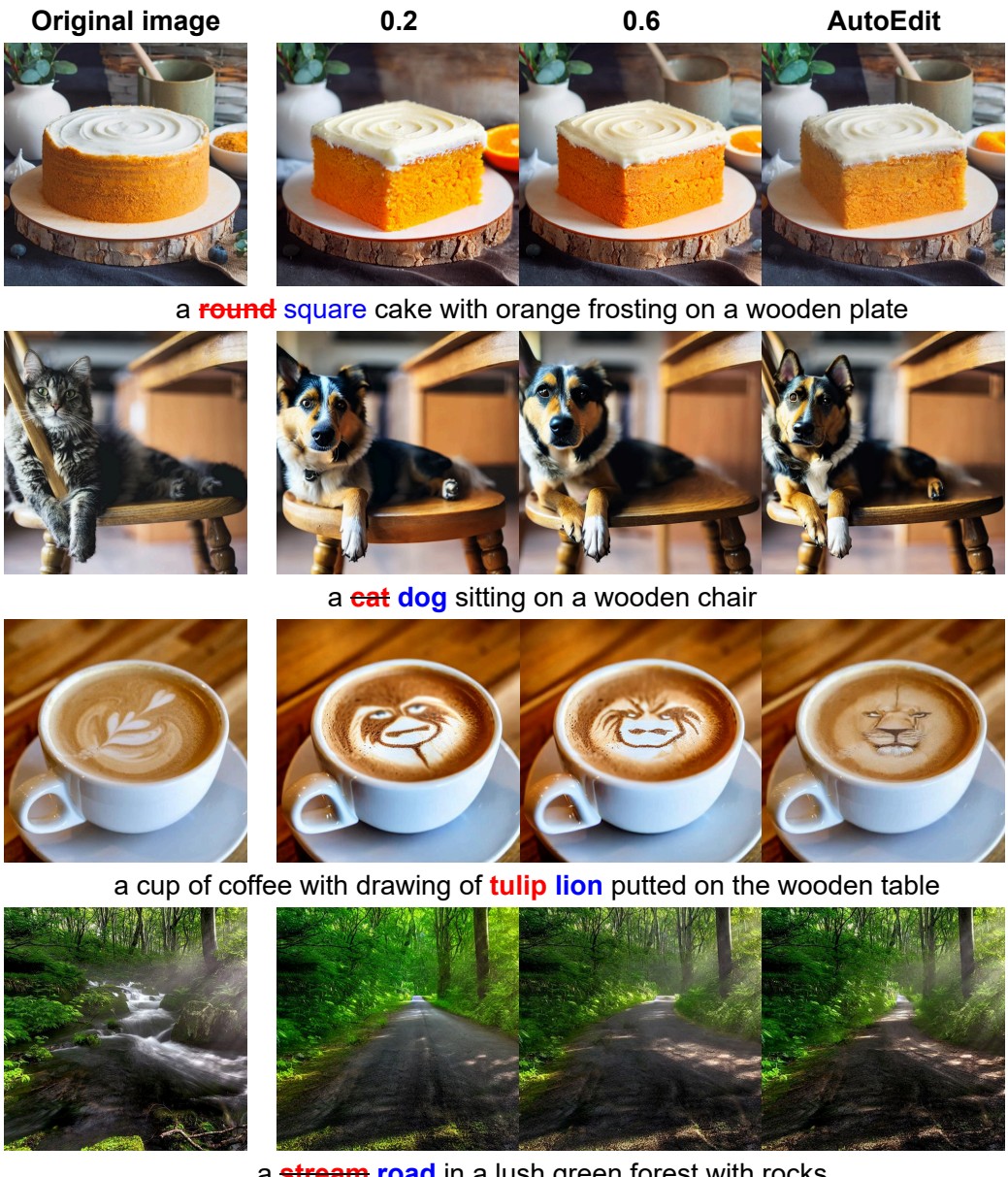

| **Original image** | **0.2** | **0.6** | **AutoEdit** |

a ~~round~~ square cake with orange frosting on a wooden plate

a ~~cat~~ **dog** sitting on a wooden chair

a cup of coffee with drawing of ~~**tulip**~~ **lion** putted on the wooden table

a ~~**stream**~~ **road** in a lush green forest with rocks

Figure 15: Compare P2P with different values of cross attention ratio. AutoEdit can be on par or better in terms of editing results without needing to tune the hyperparameters

