# OpenReview forum: "AutoEdit: Automatic Hyperparameter Tuning for Image Editing"
_NeurIPS.cc/2025/Conference — NeurIPS 2025 poster_

### Official Review · Reviewer_MdPi · 2025-06-04

**Clarity:** 2
**Significance:** 2
**Originality:** 2
**Rating:** 4
**Confidence:** 5

**Summary:**

The author proposes AutoEdit, a method that employs reinforcement learning strategies to control hyperparameters during the editing process. This approach enhances time efficiency through Proximal Policy Optimization (PPO) while maintaining optimal hyperparameter configurations.

**Questions:**

1. The authors employ DDIM inversion, which inherently introduces errors. However, the paper does not address how these errors are mitigated or their impact on editing quality.
2. The study only considers two hyperparameters—inversion timesteps and attention-replacement ratios—while omitting other critical ones, such as the Classifier-Free Guidance (CFG) scale (typically defaulted to 7.5).
3. The claimed "best editing" results lack clear evaluation metrics in Figure 1. For instance, when editing the bird image at inversion timestep=40, the bird’s posture shifts, and the branches are overly edited. Subjectively, the t=30 result appears visually superior, yet the paper does not justify the preference for t=40.

**Ethical Concerns:**

["NO or VERY MINOR ethics concerns only"]

**Final Justification:**

The author solved my main problem, and I raised the score to 4

**Limitations:**

yes

**Paper Formatting Concerns:**

There is no paper formatting concern.

**Quality:**

2

**Strengths And Weaknesses:**

Strengths:
1. The authors present an innovative approach by adapting reinforcement learning (RL) to hyperparameter selection in image editing—a relatively unexplored research direction.
2. As demonstrated in Table 1, the proposed framework exhibits strong generalizability and can be extended to diverse image editing methods.
3. Rigorous experiments are conducted, with extensive results provided in the Supplementary Material to validate the method.

Weaknesses:
1. The authors declare r as the inversion timestep in Line 120, but Figure 1 labels it as t, causing confusion. This discrepancy is not a typo, as Line 124 explicitly declares "t > r or t <= r." Additionally, the symbol definition for the attention replacement ratio remains ambiguous.
2. The paper states that an inversion timestep of 40 yields optimal results for bird image editing (Figure 1). However, the edited image visibly deviates from the original at this timestep, undermining the claim.
3. A major weakness is the absence of image reconstruction results. Empirical tests with DDIM-inversion-dependent methods (e.g., P2P, PnP) reveal that faithful reconstruction of the original image is often unattainable—a critical weakness unaddressed in this work.
4. Some minor Issues: (1). "$\bar{\alpha_t}$ decreasing to 0"(Line 90) should be corrected to "$\bar{\alpha_t}$ decreasing from 1 to 0." (2). The paper should introduce PPO in Preliminaries, a foundational RL method used in this work. (3). The term $R_{edit}$ is redundant, as it directly uses CLIP scores without modification. Labeling it as a distinct reward function may mislead readers into assuming novelty.

---

> ### Author Rebuttal · Authors · 2025-07-31
>
> Thank you for your valuable review. We will respond to your concerns as follows:
>
> **Weakness 1**: Notation confusion
>
> Thank you for your detailed review. In Figure 1, the inversion timestep should be replaced by $r$ in DDPM Inversion; however, $r$ is cross cross-attention ratio in the P2P method. We will correct Figure 1 and provide a clearer notation in the revised paper.
>
> **Weakness 2**: Suboptimal timestep in the Fig. 1
>
> Thank you for your valuable feedback. In editing tasks, the perception of the "best" edited image can be subjective and may vary from person to person. We selected that particular image as the optimal result based on a user survey, in which the majority of participants preferred it over the image generated at timestep t=30.
>
> **Weakness 3**: Reconstruction problem in DDIM Inversion:
>
> Our proposed framework does not aim to solve the reconstruction error problem inherent to inversion methods, as this is closely tied to the design of the inversion process itself. Instead, we focus on enhancing the quality of edited images within a given inversion-based editing method by selecting optimal hyperparameters. Consequently, in the case of DDIM-Inversion editing, our approach achieves improved background preservation performance (see Table 1).
>
> **Weakness 4**: Minor issues
>
> Thank you for your detailed review. We will address the issue in line 90 and include the PPO preliminaries in the revised version of the paper. Additionally, we will revise the redundant terms to improve clarity and reduce potential confusion for readers.
>
> **Question 1**: The authors employ DDIM inversion, which inherently introduces errors. However, the paper does not address how these errors are mitigated or their impact on editing quality.
>
> While DDIM inversion does introduce reconstruction errors, selecting a more suitable editing start time $r$ for each specific case can help better preserve background information while still achieving high editing quality. Our reinforcement learning framework assists in automatically selecting the optimal hyperparameters to effectively balance the trade-off between background reconstruction and prompt alignment.
>
> **Question 2**: The study only considers two hyperparameters—inversion timesteps and attention-replacement ratios—while omitting other critical ones, such as the Classifier-Free Guidance (CFG) scale (typically defaulted to 7.5):
>
> Thank you for your feedback. We also conducted an experiment using AutoEdit to adaptively search for the optimal CFG scale at each timestep. The results are presented in the following table, with the baseline's default CFG scale set to 7.5.
>
> |    Base model   | Editing method | PSNR | SSIM |  MSE | LPIPS | CLIP (edited) | CLIP (whole) | LLM metrics |
> |:---------------:|:--------------:|:----:|:----:|:----:|:-----:|:-------------:|:------------:|:-----------:|
> | SDXL            | DDPM Inversion | 26.1 | 89.7 | 30.5 | 66.2  | **23.4**          | **27.1**         | 1.04        |
> | SDXL + AutoEdit | DDPM Inversion | **29.9** | **92.2** | **15.1** | **26.3**  | 22.6          | 26.4         | **1.15**        |
>
> Experimental results show that our method significantly improves background preservation compared to the baseline, while only sacrificing a small amount of CLIP score. Moreover, our proposed AutoEdit framework boosts LLM-based evaluation metrics—where editing performance is assessed through the judgment of a large language model on the edited images—demonstrating the effectiveness of AutoEdit in searching for optimal editing hyperparameters.
>
> **Question 3**: The claimed "best editing" results lack clear evaluation metrics in Figure 1. For instance, when editing the bird image at inversion timestep=40, the bird’s posture shifts, and the branches are overly edited. Subjectively, the t=30 result appears visually superior, yet the paper does not justify the preference for t=40.
>
> Please refer to the answer to weakness 2.

---

> > ### Comment · Reviewer_MdPi · 2025-08-01
> > **The responses cannot solve my doubts**
> >
> > Thanks for the author's response. The author's response is too general and vague.
> > 1. Regarding the response to Weakness 2, the author points out that 'the' best 'edited image can be subject and may vary from person to person'. But there are no indicators that can measure this standard. Even collecting subjective opinions from participants through questionnaires can provide some reference value.
> > 2. Regarding Weakness 3, the author points out that they do not aim to solve the reconstruction error problem. However, the purpose of testing the effectiveness of image reconstruction results is to verify the degree of preservation of areas that do not require editing.

---

> > > ### Author Response · Authors · 2025-08-01
> > >
> > > Thank you for your thorough review. We would like to provide more details to answer your concerns.
> > >
> > > **Weakness 2**:
> > >
> > > We acknowledge that there is no definitive metric for determining the "best" image. However, in Figure 1, our intention was not to suggest that the image at t=40 is the best, but rather to illustrate that different images may have different preferred t values. To ensure a degree of objectivity, the selection for Figure 1 was based on a user study involving 20 participants. Each participant was asked to choose their preferred image, based on prompt alignment and background preservation. In this survey, 13 out of 20 participants preferred the edited “bird” image with t=40. We acknowledge that this selection may be biased, as users tend to prefer edited images that clearly satisfy the editing instruction (e.g., the bird appearing red at t=40), rather than subtler edits(as seen at t=30). We will clarify this detail in the caption of Figure 1 to help readers better understand the rationale behind the selection.
> > >
> > > **Weakness 3**:
> > >
> > >  We would like to clarify that common image editing methods typically consist of two phases: (1) inverting the original image to a noisy latent representation, and (2) performing the editing operation, which involves denoising guided by the edit prompt and manipulating the attention map. We acknowledge that both the inversion process and the careful control of hyperparameters that govern the editing operation can significantly influence the quality of the final edited image.
> > >
> > > In the DDIM Inversion baseline used in our paper, we report the reconstruction error of the DDIM Inversion algorithm in the table below. As shown, the mean squared error (MSE) for both the unedited regions and the entire image is relatively low. However, after applying the editing operation, the reconstruction error increases significantly to 103.95. This suggests that the editing operation has a substantial impact on the background preservation. Our proposed method, AutoEdit, specifically addresses the challenge of selecting hyperparameters that directly influence the editing operation. We present the MSE results of AutoEdit alongside the DDIM baseline in the table below, demonstrating that AutoEdit effectively improves background preservation during the editing phase.
> > >
> > > |                 | MSE (unedit part) | MSE (all) |
> > > |-----------------|-------------------|-----------|
> > > | DDIM            | 3.02              | 4.86      |
> > > | DDIM (baseline) | 103.95            | _         |
> > > | DDIM + AutoEdit | 52.94             | _         |
> > >
> > > For P2P and DDPM Inversion, we adopt the DDPM-Inversion method. This inversion approach is designed to produce negligible reconstruction error. In the PnP setting, we use the Direct Inversion technique proposed in the original paper, which introduces an effective editing method that achieves even better background preservation compared to DDIM Inversion (refer to Table 1 of the main paper). In these cases, our goal is to address the challenge of hyperparameter selection, which plays a critical role in preserving background content during the editing process.
> > >
> > > As shown in Table 1 of the main paper, AutoEdit achieves improved background preservation by selecting more effective hyperparameters during the editing process.
> > >
> > > We hope that our response will address your concern. If you have any questions, we are happy to clarify your concerns.

---

> > > > ### Author Response · Authors · 2025-08-04
> > > >
> > > > We sincerely thank you for your thorough review, which is insightful and helpful to our paper. We hope our answer could resolve your concern. If you have no more concerns, we hope that you could consider increasing the score.

---

> > > > > ### Author Response · Authors · 2025-08-07
> > > > >
> > > > > We hope that our response addresses your concerns. If you have any other concerns, please let us know.

---

> > > > > > ### Author Response · Authors · 2025-08-08
> > > > > >
> > > > > > As there is less than one day remaining for the author–reviewer discussion, we would greatly appreciate it if you could let us know of any further concerns. We are happy to provide additional clarification if needed. If there are no further concerns, we kindly hope you might reconsider your score. Thank you very much for your thorough review and for your valuable suggestions during the rebuttal, which have helped us improve and refine our paper.

---

### Official Review · Reviewer_opyq · 2025-06-22

**Clarity:** 2
**Significance:** 2
**Originality:** 3
**Rating:** 3
**Confidence:** 4

**Summary:**

This paper addresses the inefficiency in current text-guided image editing methods using diffusion models, which require brute-force tuning of multiple hyperparameters. The authors propose a reinforcement learning framework that models hyperparameter selection as a sequential decision-making process during the diffusion denoising steps. By formulating this as a Markov Decision Process and applying proximal policy optimization, their method dynamically adjusts hyperparameters based on editing objectives. Experimental results show that the approach significantly reduces computational cost and search time compared to traditional methods, making diffusion-based image editing more practical for real-world use.

**Questions:**

Can the method be applied to other models like DiT-based image generation models, since Stable Diffusion 1.4 is limited in image generation quality? If so, please include more results.

The baseline methods are a little bit old. Please include more new works.

**Ethical Concerns:**

["NO or VERY MINOR ethics concerns only"]

**Final Justification:**

After reading the rebuttal from the authors, there are some issues that have not been addressed, especially the motivation, the lack of related works and comparative experiments. Hence, I lean towards maintaining my original rating.

**Limitations:**

Please see the weaknesses and questions.

**Paper Formatting Concerns:**

No concerns

**Quality:**

2

**Strengths And Weaknesses:**

Pros:
1. The paper presents an interesting formulation of the problem and perspective.
2. The writing is clear and easy to follow.
3. The results seem promising in some cases.

Cons:
1. The authors' claim that “Assuming each $h^k$ can take on $N$ possible values, a brute-force search for the optimal configuration incurs a computational complexity of $O(TN^K)$, which is impractical in real-world scenarios." (line 45) seems to exaggerate the difficulty of the problem intentionally. Actually, for most of the methods such as prompt-to-prompt, the hyperparameters can only be selected within a small range rather than the whole range. Hence, the claim "complexity of $O(TN^K)$" is doubtful in real-world use.

2. The authors define the reward function as two essential editing criteria: prompt alignment and background preservation. However, some editing tasks should follow more than just these two basic criteria, such as style transfer, viewpoint change, etc. The authors should think more comprehensively.

3. The authors' method is based on training-free methods, which require a large amount of work selecting hyperparameters. However, many works are pretrained on large-scale data resources and are becoming the dominant direction. The authors should discuss and compare them.

4. Using the CLIP score as the reward function is not reliable, as many prior works have revealed its weak usage in image editing tasks.

---

> ### Author Rebuttal · Authors · 2025-07-31
>
> Thank you for your valuable review. We will respond to your concerns as follows:
>
> **Weakness 1**: Concern about the limited search space
>
> We acknowledge that hyperparameters can be selected from a predefined range. However, for different images, it is still necessary to identify the optimal value within that range. When dealing with a large number of images, this selection process can become time-consuming, even within a constrained range. Moreover, the editing results can vary significantly depending on the specific values chosen within that range. The complexity analysis is based solely on theoretical assumptions. Empirically, we also provide an experiment (see Table 4 in the main paper), which shows that, in practice, our proposed AutoEdit method is approximately equivalent to just three rounds of trial-and-error using the conventional approach.
>
> **Weakness 2**: Concern about the reward function for several editing tasks
>
> - For style transfer editing, we omit the background preservation terms in our reward computation, consistent with the evaluation protocol of the Pie-Bench dataset.
> - Regarding viewpoint changes, this task is more akin to 3D reconstruction than image editing, and thus falls outside the scope of our work. In most viewpoint-change scenarios, 3D-aware diffusion models or score distillation methods are preferred. In such cases, reinforcement learning could potentially be integrated into the training process to produce more consistent viewpoint generation.
>
> **Weakness 3**: AutoEdit with training-based approaches:
>
> Training-based editing methods can be regarded as specialized generation pipelines that condition on both the original image and the instruction prompt, rather than relying solely on text-to-image models. For example, in [1] and [2], our reinforcement learning framework can be employed to optimize the classifier-free guidance (CFG) coefficient during the denoising process.
>
> **Weakness 4**: CLIP score function is not a reliable measure
>
> We acknowledge that relying solely on the CLIP score may be insufficient for capturing fine-grained edits, such as changes in small or detailed regions (e.g., eye color). To address this limitation, we incorporate large language model (LLM)-based evaluation to assess editing quality. Specifically, we employ a multi-step approach using GPT-4o to generate and compute rewards as follows:
>
> - **Step 1**: Given an editing instruction (e.g., "change the eye color to blue"), we prompt GPT-4o to generate a corresponding question (e.g., yes/no or multiple-choice) along with a ground truth answer. For example: Question: "What is the color of the eye?" Choices: ["black", "blue"]; Answer: "blue".
> - **Step 2**: To compute the reward, we query the LLM for both foreground editing and background preservation. For foreground editing, we use the question and ground truth answer from Step 1 and require GPT-4o to identify the correct choice. For background preservation, we ask the LLM to compare the edited image with the original and assign a score (0, 0.5, or 1) based on the quality of background consistency. The total reward is the sum of these two components.
>
> We conduct experiments using SDXL with DDPM-Inversion, tuning the inversion timestep based on the newly introduced LLM-based reward. The following table presents a comparison between our original CLIP-based reward and the new LLM-based reward, with the LLM score being the GPT-4o evaluation score, which is similar to the computation of the reward function:
>
> |       Method      | PSNR | SSIM |  MSE | LPIPS | CLIP Score (edited) | CLIP score (Whole) | LLM score |
> |:-----------------:|:----:|:----:|:----:|:-----:|:-------------------:|:------------------:|:---------:|
> | SDXL              | 26.1 | 89.7 | 30.5 | 66.2  | **23.4**                | **27.1**               | 1.04      |
> | SDXL + AutoEdit   | **27.8** |**90.5** | **20.4** | **53.5**  | 23.1                | 26.9               | **1.15**      |
> | SD 1.4            | 22.6 | 78.9 | 53.3 | 67.6  | **23.0**                |**26.2**               | 1.01      |
> | SD 1.4 + AutoEdit | **27.2** |**85.2** | **31.1** | **50.5**  | 22.5                | 25.8               | **1.10**      |
>
> Experimental results show that our method consistently enhances the performance of baseline approaches in terms of background preservation metrics, while exhibiting only a slight decline in CLIP score. We argue that this minor reduction does not compromise the overall editing capability. Furthermore, evaluations using GPT-4o indicate that our proposed method effectively improves the editing quality over the baseline.
>
> **Question 1**: Can the method be applied to other models like DiT-based image generation models, since Stable Diffusion 1.4 is limited in image generation quality?
>
> Thank you for your thorough review. Our framework can be extended to DiT-based image generation models. Specifically, we conduct an experiment using [3], where Auto-Edit is employed to automatically select the optimal attention injection timestep. The results are presented in the following table:
>
> |         Method         | PSNR | SSIM | CLIP Score (edited) | CLIP score (whole) | LLM score |
> |:----------------------:|:----:|:----:|:-------------------:|:------------------:|:---------:|
> | Taming Flow            | 23.4 | 81.5 | **22.9**                |**26.0**              | 1.02      |
> | Taming Flow + AutoEdit | **26.1** | **85.4** | 22.7                | 25.6               | **1.13**      |
>
> Our proposed AutoEdit can improve the performance of [3] on background preservation and LLM score, where LLM score is the evaluation of the editing quality provided by a GPT-4o model with a slight decrease in terms of CLIP score. However, we argue that the small decrease of the CLIP score does not affect the editing capability of AutoEdit.
>
> [1]InstructPix2Pix: Learning to Follow Image Editing Instructions
>
> [2] SmartEdit: Exploring Complex Instruction-based Image Editing with Multimodal Large Language Models
>
> [3]Taming rectified flow for inversion and editing, ICML 2025.

---

> > ### Author Response · Authors · 2025-08-04
> >
> > We sincerely thank you for your thorough review, which is insightful and helpful to our paper. We hope our answer could resolve your concern. If you have no more concerns, we hope that you could consider increasing the score.

---

> ### Comment · Reviewer_opyq · 2025-08-06
> **Reply to Authors**
>
> Thanks for the effort the authors have made. However, my major concerns were not addressed.
>
> Firstly, the claim of complexity $O(TN^k)$ is not practical and cannot be justified by “When dealing with a large number of images, this selection process can become time-consuming” as $O(TN^k)$ is for one image.
>
> Secondly, the viewpoint change, or size change, action change, belong to the image editing field, as prior works have explored (e.g., https://arxiv.org/abs/2309.10556, https://arxiv.org/abs/2304.08465). The authors' reply of "...falls outside the scope of our work" is doubtful.
>
> Thirdly, comparison results with training-based approaches are missing.
>
> Lastly, the proposed LLM score is underexplored. There are also many similar evaluations (https://arxiv.org/abs/2507.16193), while the authors did not cite or investigate them.
>
> Overall, this paper still needs further revision.

---

> > ### Author Response · Authors · 2025-08-07
> >
> > Thank you for your thorough reviews. We would like to clarify your concerns:
> >
> > 1. **the claim of complexity $O(TN^k)$ is not practical and cannot be justified by “When dealing with a large number of images, this selection process can become time-consuming” as $O(TN^k)$ is for one image.**
> >
> > We acknowledge that the claim “complexity $O(TN^k)$ is not practical” cannot be fully justified by the large number of images alone where each image may require a different optimal hyperparameter for editing. However, the key message is that in scenarios involving automatic editing of many images, AutoEdit alleviates the burden of manually selecting hyperparameters for each image individually.
> >
> > While it is true that optimal hyperparameters can often be defined within a specific range, the optimal value may vary across different diffusion timesteps t. Selecting hyperparameters for each timestep individually is a labor-intensive process. For instance, during the generation process, the CFG scale has varying effects across timesteps [1], and tuning CFG per timestep has been shown to improve generation quality.
> >
> > The theoretical complexity $O(TN^k)$ is presented to illustrate the computational burden of manual hyperparameter search. To complement this, Table 5 in the main paper provides a practical comparison between manual hyperparameter tuning (conducted by humans) and our AutoEdit framework. The results show that AutoEdit achieves comparable editing quality to that of three rounds of manual hyperparameter tuning, demonstrating its effectiveness and efficiency.
> >
> > [1]:  Applying Guidance in a Limited Interval Improves Sample and Distribution Quality in Diffusion Models
> >
> > 2. **the viewpoint change, or size change, action change, belong to the image editing field, as prior works have explored (e.g., https://arxiv.org/abs/2309.10556, https://arxiv.org/abs/2304.08465). The authors' reply of "...falls outside the scope of our work" is doubtful.**
> >
> > Thank you for your clarification. We initially misunderstood your comment as referring specifically to 3D novel view synthesis involving viewpoint changes. After reading the paper you suggested, we noticed that it also includes side-view generation and action editing. However, for side-view generation, MasaCtrl does not achieve the same level of consistency as typical novel view synthesis methods.
> >
> > In Table 1 of the main paper, we have already included the results for MasaCtrl, and MasaCtrl+AutoEdit outperforms the baseline in both background preservation and prompt alignment. Additionally, we would like to provide the editing results for the pose-changing setting on the PIE-Bench dataset.
> >
> > | Method              | Base model | PSNR | SSIM | MSE  | LPIPS | CLIP edit | CLIP full |
> > |---------------------|------------|------|------|------|-------|-----------|-----------|
> > | DDPM Inv            | SDXL       | 26.6 | 88.6 | 34.0 | 66.1  | 23.0      | 27.4      |
> > | DDPM Inv + AutoEdit | SDXL       | **29.3** | **90.8** | **19.1** | **47.5**  | **23.5**      | **27.8**      |
> > | MasaCtrl            | SD 1.4     | 22.5 | 78.3 | 83.1 | 105.6 | 22.2      | 26.2      |
> > | MasaCtrl + AutoEdit | SD 1.4     | **23.2** | **80.1** | **59.6** | **90.7**  | **22.8**      | **26.7**      |
> >
> > The table above shows that AutoEdit achieves superior editing quality compared to the baseline, indicating its effectiveness in pose editing, closely related to tasks such as action change and viewpoint change. We also experimented with MasaCtrl+AutoEdit using side-view and action-editing prompts similar to those in the MasaCtrl examples, and observed that MasaCtrl+AutoEdit produces better qualitative results. However, due to rebuttal policy, we are unable to include the images here and will update them in the revised version of the paper.
> >
> > Additionally, in cases that primarily focus on novel view synthesis, we could collect a dedicated dataset for this task and train AutoEdit on a multi-view diffusion model. The reward function can be designed to measure the consistency between the synthesized novel views and the ground truth using metrics such as PSNR, SSIM, and LPIPS.

---

> ### Author Response · Authors · 2025-08-07
>
> 3. **Comparison results with training-based approaches are missing.**
>
> We provide the additional result of Autoedit on instruction-based editing which is the training-based editing methods, including Instruct Pix2Pix [2] and UltraEdit [3] to choose the CFG for each timestep.
>
> | Method                      | Base model    | PSNR | SSIM | MSE   | LPIPS | CLIP edit | CLIP full |
> |-----------------------------|---------------|------|------|-------|-------|-----------|-----------|
> | Instruct Pix2Pix            | SD 1.5 (Unet) | 20.8 | 76.4 | 226.8 | 157.3 | 22.1      | 24.5      |
> | Instruct Pix2Pix + AutoEdit | SD 1.5 (Unet) | **22.2** | **78.5** | **181.4** | **132.8** | **22.3**      | **24.7**      |
> | Ultra Edit                  | SD 3 (MM-DiT) | 26.5 | 84.7 | 46.7  | 75.8  | 22.4      | 25.6      |
> | Ultra Edit + AutoEdit       | SD 3 (MM-DiT) | **27.3** | **86.2** | **37.6**  | **64.9**  | **22.6**      | **25.7**      |
>
> AutoEdit improves background preservation and prompt alignment over the above baseline methods. Experimental results demonstrate that AutoEdit is applicable to training-based approaches, such as instruction-based editing methods, highlighting its generalizability across different editing techniques and model architectures.
>
> [2] Brooks et al. , InstructPix2Pix Learning to Follow Image Editing Instructions, CVPR 2023
>
> [3] Zhao, Haozhe, et al. "Ultraedit: Instruction-based fine-grained image editing at scale." Advances in Neural Information Processing Systems 37 (2024): 3058-3093.
>
> 4. **the proposed LLM score is underexplored. There are also many similar evaluations (https://arxiv.org/abs/2507.16193), while the authors did not cite or investigate them.**
>
> The suggested paper was published on arXiv on July 22, 2025, which is after the NeurIPS submission deadline. Therefore, we were not aware of it at the time of submission. However, we appreciate the suggestion and will include a reference to this paper along with a discussion of its relevance in the revised version after the rebuttal phase.
>
> Regarding the LLM score, a recent paper [4] has explored the use of LLM-based evaluation for image editing tasks. We would like to emphasize that our main contribution lies in proposing a reinforcement learning framework with a flexible reward function, such as the LLM score. Following your suggestion, we will include a more comprehensive reference and discussion of related work in the revised paper.
>
> [4] I2EBench: A Comprehensive Benchmark for Instruction-based Image Editing
>
> We hope that our response addresses your concerns. If you have any other concerns, please let us know.

---

> > ### Author Response · Authors · 2025-08-08
> >
> > As there is less than one day remaining for the author–reviewer discussion, we would greatly appreciate it if you could let us know of any further concerns. We are happy to provide additional clarification if needed. If there are no further concerns, we kindly hope you might reconsider your score. Thank you very much for your thorough review and for your valuable suggestions during the rebuttal, which have helped us improve and refine our paper.

---

### Official Review · Reviewer_j63C · 2025-06-26

**Clarity:** 3
**Significance:** 3
**Originality:** 3
**Rating:** 4
**Confidence:** 4

**Summary:**

This paper proposes AutoEdit, which can automatically searches the optimal hyperparameters in image editing. This is achieved by establishment of a Markov Decision Process which intergrates serveral editing objectives into a reward function. Extensive experiments illustrate the effectiveness of the proposed method.

**Questions:**

See weakness.

**Ethical Concerns:**

["NO or VERY MINOR ethics concerns only"]

**Final Justification:**

The rebuttal addresses my concerns. As a result, I would like to maintain the score as borderline accept.

**Limitations:**

yes

**Quality:**

3

**Strengths And Weaknesses:**

Strength:
- The motivation of this paper is clear. Hyperparameter-tuning is a long-last challenging problem in the image editing tasks. Designing methods to alleviate this problem is meaningful.
- The proposed method is novel and interesting, treating the problem of optimizing hyperparameters as a RL task.
- The experiment result seems impressive.

Weakness:
- For each specified image-editing method, AutoEdit requires employing PPO to train the model, which may lead to a substantial computational resource consumption. Could you provide the detailed information of the resources and times consumed during the training process of AutoEdit?
- The Background preservation is considered in the reward function. If I understand correctly, this can result in a reduction of global editing capability (e.g., editing the style of the image or changing the season of a image from Spring to Winter).
- All of the experiments are conducted on the U-Net-based method (such as P2P and PnP), these approaches are now somewhat outdated. Rectified flow–based DiT architectures have recently achieved state-of-the-art performance in image generation (such as FLUX), and corresponding rectified-flow image-editing methods have been proposed (such as RF-Solver[1] and FireFlow[2]). Could you integrate your method into the RF-DiT model to demonstrate its generalizability?


[1]. Taming rectified flow for inversion and editing, ICML 2025.

[2]. FireFlow: Fast Inversion of Rectified Flow for Image Semantic Editing, ICML 2025.

---

> ### Author Rebuttal · Authors · 2025-07-31
>
> Thank you for your valuable review. We will respond to your concerns as follows:
>
> **Weakness 1**: Training time of AutoEdit
>
> For all experiments in the main paper, the total training time was approximately 10 hours on NVIDIA A6000 GPUs, covering both Stage 1 and Stage 2 training.
>
> For the additional experiments with SDXL conducted during the rebuttal phase, we use NVIDIA A100 GPUs with a total training time of approximately 12 hours.
>
> **Weakness 2**: Concern about the mask for background preservation for the global editing task (style transfer)
>
> For the global editing task, we only employ the CLIP score as the reward function, which is similar to the evaluation protocol of the PieBench dataset. We present the results of the style transfer task in the following table:
>
> | Base model |    Editing Method   | CLIP Score |
> |:----------:|:-------------------:|:----------:|
> | SDXL       | DDPM Inv            | 26.5       |
> | SDXL       | DDPM Inv + AutoEdit | **28.3**       |
> | SD 1.5     | P2P                 | 25.5       |
> | SD 1.5     | P2P + AutoEdit      | **26.7**       |
>
> From this table, our proposed AutoEdit can improve the editing results on the Style Transfer task compared to the baseline model.
>
> **Weakness 3**: Lack of editing on recent methods
>
> Thank you for your suggestion. Due to the time constraint, we are only able to perform AutoEdit for the Rf-solver. In this setting, we leverage our RL framework to determine the optimal attention injection timestep. The performance of Auto-Edit under this configuration is presented in the following table:
>
> |         Method         | PSNR | SSIM | CLIP Score (edited) | CLIP score (whole) | LLM score |
> |:----------------------:|:----:|:----:|:-------------------:|:------------------:|:---------:|
> | Taming Flow            | 23.4 | 81.5 | **22.9**                | **26.0**               | 1.02      |
> | Taming Flow + AutoEdit | **26.1** | **85.4** | 22.7                | 25.6               | **1.13**      |
>
> Our proposed AutoEdit can improve the performance of Taming Flow on background preservation and LLM score, where LLM score is the evaluation of the editing quality provided by a GPT-4o model with a slight decrease in terms of CLIP score. However, we argue that the small decrease of CLIP score does not affect the  editing capability of AutoEdit.

---

> ### Comment · Reviewer_j63C · 2025-08-04
>
> Thanks for author's response. The topic of this paper is interesting and still under-explored by previous works. I would maintain my score as weak accept.

---

> > ### Author Response · Authors · 2025-08-04
> >
> > We appreciate your time for thoroughly reviewing our paper. Thank you for considering our rebuttal.

---

### Official Review · Reviewer_qdGc · 2025-06-29

**Clarity:** 3
**Significance:** 2
**Originality:** 3
**Rating:** 4
**Confidence:** 5

**Summary:**

The authors introduce a reinforcement learning (RL) framework designed to automatically tune hyperparameters for diffusion-based image editing. The paper frames the challenge of finding the best settings for parameters like inversion timesteps and attention modifications as a sequential decision-making task. By establishing a Markov Decision Process within the diffusion denoising process, the proposed method uses a policy model to dynamically adjust hyperparameters at each step, optimizing the losses obtained by edit quality and background preservation. This approach finds near-optimal configurations in a single pass, avoiding the costly trial-and-error of traditional brute-force searches.

**Questions:**

- How can it be integrated into other image editing tasks (such as shape changes, local attribute editing, etc.)
- What about its performance on various models, such as different versions of SD, and how are them compared to more recent baselines (such as TF-ICON , IMPRINT, MIGC)?


TF-ICON: https://github.com/Shilin-LU/TF-ICON

IMPRINT: https://openaccess.thecvf.com/content/CVPR2024/papers/Song_IMPRINT_Generative_Object_Compositing_by_Learning_Identity-Preserving_Representation_CVPR_2024_paper.pdf

MIGC: https://openaccess.thecvf.com/content/CVPR2024/papers/Zhou_MIGC_Multi-Instance_Generation_Controller_for_Text-to-Image_Synthesis_CVPR_2024_paper.pdf

**Ethical Concerns:**

["NO or VERY MINOR ethics concerns only"]

**Final Justification:**

Rebuttal provides extra results regarding the method's generalization issues, which was my primary concern, therefore, i raised my score to 4.

**Limitations:**

yes.

**Paper Formatting Concerns:**

no issues

**Quality:**

2

**Strengths And Weaknesses:**

Strengths:
- The hypothesis and problem definition is clear, the optimal hyperparameters for image editing are highly dependent on the specific image and edit, making manual tuning inefficient. It systematically formulates this issue by identifying the exponential complexity of brute-force search (and proposes an RL-based solution that reduces this complexity to a linear.

- The proposed approach is time-efficient, reducing the O(TN^K)) time complexity into O(T).

- The motivation for the work is strong and clearly described. The authors address the computational costs and usability barriers associated with the manual, trial-and-error hyperparameter searches required by existing image editing methods. By aiming to automate this process, the work is well-motivated to improve the practical deployment and accessibility of advanced image editing tools.

Weaknesses:
- The proposed RL formulation is dependent on a predefined edit mask M. The reward function is explicitly designed to calculate background preservation on the unmasked region, so it limits the models that are compatible with the approach. Is there any extension for non-masked based editings? (1-M) and prompt alignment on the masked region M relies on a mask inherently limits the framework's applicability to editing tasks that can be clearly segmented, such as object replacement or inpainting, while excluding global edits like style changes that do not have a discrete mask.

- The framework's reliance on a segmentation mask may also limit its effectiveness for very fine-grained attribute edits (e.g., changing a person's eye color). Generating a precise mask for such a small, detailed region can be challenging, and the CLIP-based reward signal calculated on that tiny area may not be robust enough to guide the policy effectively. The paper's methodology seems better suited for object-level edits where clear segmentation is more feasible.

- The experiments, while applying AutoEdit to several editing methods, implement all of them on a single underlying diffusion model: Stable Diffusion 1.4. This makes it difficult to assess how well the AutoEdit framework generalizes to other diffusion models, such as SDXL or other versions of SD architectures. A more comprehensive analysis would demonstrate its effectiveness across a variety of base models.

- The comparison is made against a limited set of baselines, with the most recent being MasaCtrl, which was published in 2023. The field of image editing has advanced since, with newer and more powerful models being released (see Questions). Failing to compare against more recent state-of-the-art methods limits the conclusions that can be drawn about AutoEdit's performance relative to the current landscape.

---

> ### Author Rebuttal · Authors · 2025-07-31
>
> Thank you for your valuable review. We will respond to your concerns as follows:
>
> **Weakness 1**: The dependent of mask $M$ in the reward computation
>
> Our training objective is based on an editing metric. For non-masked editing tasks such as style transfer, the background preservation term (MSE) is excluded from the reward computation.
>
> For global editing tasks (e.g., style transfer or style change), we use only the prompt alignment metric as the reward function. This choice aligns with the evaluation protocol used in Pie Bench, our benchmark dataset.
>
> Additionally, we provide a comparison table highlighting our method outperforms the baseline in the style transfer task.
> | Base model |    Editing Method   | CLIP Score |
> |:----------:|:-------------------:|:----------:|
> | SDXL       | DDPM Inv            | 26.5       |
> | SDXL       | DDPM Inv + AutoEdit | **28.3**       |
> | SD 1.5     | P2P                 | 25.5       |
> | SD 1.5     | P2P + AutoEdit      | **26.7**       |
>
> The experiment demonstrates the effectiveness of AutoEdit on the style transfer task.
>
> **Weakness 2**: The framework's reliance on the segmentation mask limits its effectiveness for very fine-grained attribute edits
>
> The performance of image editing depends heavily on both the reward function and the capabilities of the base model. In this work, we propose a general reinforcement learning (RL) framework for hyperparameter selection in editing tasks. This framework is flexible and can accommodate various reward functions.
>
> We acknowledge that relying solely on the CLIP score within the masked region may be insufficient for capturing fine-grained edits, such as changes in small or detailed regions (e.g., eye color). To address this limitation, we incorporate large language model (LLM)-based evaluation to assess editing quality. Specifically, we employ a multi-step approach using GPT-4o to generate and compute rewards as follows:
>
> - **Step 1**: Given an editing instruction (e.g., "change the eye color to blue"), we prompt GPT-4o to generate a corresponding question (e.g., yes/no or multiple-choice) along with a ground truth answer. For example: Question: "What is the color of the eye?" Choices: ["black", "blue"]; Answer: "blue".
> - **Step 2**:  To compute the reward, we query the LLM for both foreground editing and background preservation. For foreground editing, we use the question and ground truth answer from Step 1 and require GPT-4o to identify the correct choice. For background preservation, we ask the LLM to compare the edited image with the original and assign a score (0, 0.5, or 1) based on the quality of background consistency. The total reward is the sum of these two components.
>
> We conduct experiments using SDXL with DDPM-Inversion, tuning the inversion timestep based on the newly introduced LLM-based reward. The following table presents a comparison between our original CLIP-based reward and the new LLM-based reward, with LLM score is the GPT-4o evaluation score, which is similar to the computation of the reward function.
>
> |          Base model          | Editing method | PSNR | SSIM |  MSE | LPIPS | CLIP (edited) | CLIP (whole) | LLM score |
> |:----------------------------:|:--------------:|:----:|:----:|:----:|:-----:|:-------------:|:------------:|:---------:|
> | SDXL                         | DDPM Inv       | 26.1 | 89.7 | 30.5 | 66.2  | **23.4**          | **27.1**         | 1.04      |
> | SDXL + AutoEdit              | DDPM Inv       | 27.8 | 90.5 | 20.4 | 53.5  | 23.1          | 26.9         | 1.15      |
> | SDXL + AutoEdit + LLM reward | DDPM Inv       | **29.4** | **92.2** | **19.1** | **44.1**  | 22.9          | 26.7         | **1.20**      |
>
> Our AutoEdit framework consistently improves both background preservation metrics and LLM-based judgment scores, while exhibiting only a slight decline in CLIP score. We argue that this minor decrease in CLIP score does not compromise the overall editing performance of AutoEdit.
>
> **Weakness 3**: Concerns about the SD 1.4 model:
>
> We conduct experiments using SDXL with DDPM-Inversion, tuning the inversion timestep accordingly. The table below presents a comparison between Auto-Edit and baseline methods on the SDXL base model:
>
> |       Method      | PSNR | SSIM |  MSE | LPIPS | CLIP Score (edited) | CLIP score (Whole) | LLM score |
> |:-----------------:|:----:|:----:|:----:|:-----:|:-------------------:|:------------------:|:---------:|
> | SDXL              | 26.1 | 89.7 | 30.5 | 66.2  | **23.4**                | **27.1**               | 1.04      |
> | SDXL + AutoEdit   | **27.8** | **90.5** | **20.4** | **53.5**  | 23.1                | 26.9               | **1.15**      |
> | SD 1.4            | 22.6 | 78.9 | 53.3 | 67.6  | **23.0**                | **26.2**               | 1.01      |
> | SD 1.4 + AutoEdit | **27.2** | **85.2** | **31.1** | **50.5**  | 22.5                | 25.8               | **1.10**      |
>
> Our results demonstrate that AutoEdit significantly improves background preservation quality, with only a slight reduction in CLIP score. However, we argue that this minor decrease in CLIP score does not substantially impact the overall editing quality. Furthermore, evaluations using large language models (LLMs) indicate that AutoEdit considerably enhances the editing capabilities of baseline methods.
>
> **Weakness 4**:  Concern about the limited set of editing methods.
>
> Thank you for your suggestion. The baseline you mentioned pertains to image composition tasks, which follow a different setup compared to image editing, the primary focus of our work. Nonetheless, our proposed RL framework is general and can be readily applied to diffusion-based image composition tasks for hyperparameter optimization.
>
> For example, in the TF-ICON paper, our RL framework can be parameterized to optimize the injection timesteps of the composite self-attention maps, denoted as $\tau_A$ and $\tau_B$—a set of hyperparameters explicitly defined in the method. The reward function can be constructed based on the quality of foreground integration and background preservation in the composited image.
>
> We also conduct experiments on a recent editing method [1]. In this setting, we leverage our RL framework to determine the optimal attention injection timestep. The performance of Auto-Edit under this configuration is presented in the following table.
>
> |         Method         | PSNR | SSIM | CLIP Score (edited) | CLIP score (whole) | LLM score |
> |:----------------------:|:----:|:----:|:-------------------:|:------------------:|:---------:|
> | Taming Flow            | 23.4 | 81.5 |**22.9**                | **26.0**               | 1.02      |
> | Taming Flow + AutoEdit | **26.1** | **85.4** | 22.7                | 25.6               | **1.13**      |
>
> Our experiments demonstrate that Auto-Edit enhances the editing performance of the baseline methods. Specifically, we observe improvements in the LLM-based evaluation metric and background preservation scores. While there is a slight decrease in the CLIP score, this suggests that our method effectively discovers hyperparameters that balance foreground fidelity and background consistency, resulting in overall improved editing quality.
>
> **Question 1**: How can it be integrated into other image editing tasks (such as shape changes, local attribute editing, etc.)?
>
> Our method is applicable to a wide range of editing tasks due to the flexibility of its reward function. For instance, the CLIP score can be used as a reward for style transfer tasks, while LLM-based evaluation can be employed to assess correctness in local attribute editing.
>
> It is important to note that certain challenging editing tasks may also require stronger base models and novel editing techniques. Our approach focuses on the overarching editing problem by selecting optimal hyperparameters for each editing method (see Fig. 3 and Table 1 in our main paper).
>
> **Question 2**: What about its performance on various models, such as different versions of SD, and how are they compared to more recent baselines (such as TF-ICON, IMPRINT, MIGC)?
>
> Please refer to the response to weakness 4.
>
> [1]Taming Rectified Flow for Inversion and Editing (ICML 2025).

---

> > ### Author Response · Authors · 2025-08-04
> >
> > We sincerely thank you for your thorough review, which is insightful and helpful to our paper. We hope our answer could resolve your concern. If you have no more concerns, we hope that you could consider increasing the score.

---

> > ### Comment · Reviewer_qdGc · 2025-08-06
> >
> > Thank you for the detailed rebuttal. While I appreciate the effort, my primary concern regarding generalization remains.
> >
> > The results with SDXL and Taming Flow, while welcome, are not sufficient to fully demonstrate the claimed generalization capabilities. Since the Stable Diffusion models belong to a similar architectural family, strengthening the claims would require showing results on a different type of model architecture. Furthermore, the evaluation does not include experiments with different types of editing methods.
> >
> > Because this main concern has not been fully addressed, I am leaning towards keeping my original score.

---

> > > ### Author Response · Authors · 2025-08-06
> > >
> > > Thank you for your comment. Taming Flow is based on the FLUX model, which utilizes a DiT transformer architecture combined with flow matching. In our rebuttal experiments, we demonstrate that AutoEdit achieves a strong balance between prompt alignment and background preservation for both UNet (SDXL, SD1.4) and Transformer (DiT) architecture.
> > >
> > > We would greatly appreciate it if the reviewer could suggest specific additional editing methods that may help address your concern regarding generalization.

---

> > > > ### Comment · Reviewer_qdGc · 2025-08-06
> > > >
> > > > To further validate the approach, I suggest evaluating its performance on additional editing tasks, such as instruction-based edits. A comparison with the following relevant models would be particularly insightful:
> > > >
> > > > [1] Mokady, Ron, et al. "Null-text inversion for editing real images using guided diffusion models." Proceedings of the IEEE/CVF conference on computer vision and pattern recognition. 2023.
> > > >
> > > > [2] Zhao, Haozhe, et al. "Ultraedit: Instruction-based fine-grained image editing at scale." Advances in Neural Information Processing Systems 37 (2024): 3058-3093.
> > > >
> > > > [3] Wasserman, Navve, et al. "Paint by inpaint: Learning to add image objects by removing them first." arXiv preprint arXiv:2404.18212 (2024).
> > > >
> > > > [4] Brooks et al. , InstructPix2Pix Learning to Follow Image Editing Instructions, CVPR 2023

---

> ### Author Response · Authors · 2025-08-07
>
> Due to the time constraint, we are only able to conduct the experiments using the instruction-based editing methods UltraEdit[2] and InstructPix2Pix[4] to choose the CFG for each timestep. Our proposed AutoEdit framework enables dynamic selection of CFG values across different denoising timesteps, thereby improving the overall performance of the editing methods, as shown in the table below:
>
> | Method                      | Base model    | PSNR | SSIM | MSE   | LPIPS | CLIP edit | CLIP full |
> |-----------------------------|---------------|------|------|-------|-------|-----------|-----------|
> | Instruct Pix2Pix            | SD 1.5 (Unet) | 20.8 | 76.4 | 226.8 | 157.3 | 22.1      | 24.5      |
> | Instruct Pix2Pix + AutoEdit | SD 1.5 (Unet) | **22.2** | **78.5** | **181.4** | **132.8** | **22.3**      | **24.7**      |
> | Ultra Edit                  | SD 3 (MM-DiT) | 26.5 | 84.7 | 46.7  | 75.8  | 22.4      | 25.6      |
> | Ultra Edit + AutoEdit       | SD 3 (MM-DiT) | **27.3** | **86.2** | **37.6**  | **64.9**  | **22.6**      | **25.7**      |
>
> AutoEdit improves background preservation and prompt alignment over the above baseline methods. Experimental results demonstrate that AutoEdit is applicable to training-based approaches, such as instruction-based editing methods, highlighting its generalizability across different editing techniques and model architectures.
>
> We hope that our response addresses your concerns. If you have any other concerns, please let us know.

---

> > ### Comment · Reviewer_qdGc · 2025-08-07
> >
> > I thank for the authors' efforts in providing new results, and they partially address my concerns. I will re-consider my evaluation given your rebuttal. Thanks.

---

> > > ### Author Response · Authors · 2025-08-08
> > >
> > > Thank you for your reconsideration and thorough review for our paper and rebuttal process. If there is any other concerns you might have, we are happy to provide further clarification.
> > >
> > > Thank you very much!

---

> > > > ### Author Response · Authors · 2025-08-08
> > > >
> > > > As there is less than one day remaining for the author–reviewer discussion, we would greatly appreciate it if you could let us know of any further concerns. We are happy to provide additional clarification if needed. If there are no further concerns, we kindly hope you might reconsider your score. Thank you very much for your thorough review and for your valuable suggestions during the rebuttal, which have helped us improve and refine our paper.

---

> > > > > ### Author Response · Authors · 2025-08-09
> > > > >
> > > > > We present additional results of applying AutoEdit to Paint-By-Inpaint [3] and Null-text Inversion [1], in comparison with their respective baselines. For Paint-By-Inpaint, AutoEdit is employed to search for the optimal CFG value, while for Null-text Inversion, it is used to determine the optimal attention injection timestep. The results are shown in the table below:
> > > > >
> > > > > | Method                        | PSNR | SSIM | MSE   | LPIPS | CLIP edit | CLIP full |   |
> > > > > |-------------------------------|------|------|-------|-------|-----------|-----------|---|
> > > > > | Pain-By-Inpaint               | 20.5 | 76.7 | 261.3 | 161.2 | 21.3      | 23.7      |   |
> > > > > | Pain-By-Inpaint + AutoEdit    | **21.2** | **77.8** | **227.1** | **146.3** | **21.5**      | **23.8**      |   |
> > > > > | Null-text inversion           | 23.8 | 79.9 | 64.4  | 109.8 | 22.3      | 25.9      |   |
> > > > > | Null-text inversion+ AutoEdit | **25.7** | **82.4** | **45.4**  | **82.3**  | **22.6**      | **26.3**      |   |
> > > > >
> > > > > Our proposed AutoEdit improves baseline performance in both background preservation and prompt alignment, demonstrating its generalizability across various editing methods.
> > > > >
> > > > > We hope that our provided results can completely address your concern. Thank you for considering our rebuttal

---

### Comment · Area_Chair_zCUN · 2025-08-04

Dear reviewers,

Please post your response as soon as possible, if you haven’t done it, to allow time for discussion with the authors. All reviewers should respond to the authors' rebuttal to confirm it has been read.

Thanks,

AC

---

### Decision · Program_Chairs · 2025-09-17

**Decision:**

Accept (poster)

**Comment:**

This paper presents AutoEdit, a RL framework to automate hyperparameter tuning in image editing. This method avoids slow manual tuning by treating the problem as a sequential decision task. The core idea seems novel. However, the initial submission had some weaknesses. It relied on segmentation masks, which limits its use. The experiments were also tested on older models, e.g. SD1.4.

During the discussion, reviewers raised important points. Reviewers qdGc, j63C, and opyq questioned the method's generalizability to different models and editing tasks. They also pointed out the limitations of using a CLIP-based reward function. In response, the authors provided many new experiments. They showed AutoEdit works with newer models like SDXL and DiT architectures, and on more tasks like instruction-based editing. They also introduced an LLM-based reward system. These new results convinced most reviewers. After reading all discussions, I generally agree with the reviewers and recommend accepting the paper, as the authors thoroughly addressed the reviewers’ concerns. Please incorporate the SDXL and SD3 results into the revision.